# MULTIMODAL DISTILLATION OF PROTEIN SEQUENCE, STRUCTURE, AND FUNCTION

## ABSTRACT

Proteins are the fundamental building blocks of life, carrying out essential biological functions in biology. Learning effective representations of proteins is critical for important applications like drug design and function prediction. Language models (LMs) and graph neural networks (GNNs) have shown promising performance for modeling proteins. However, multiple data modalities exist for proteins, including sequence, structure, and functional annotations. Frameworks integrating these diverse sources without large-scale pre-training remain underdeveloped. In this work, we propose ProteinSSA, a multimodal knowledge distillation framework to incorporate **Protein S**equence, **S**tructure, and Gene Ontology (GO) **A**nnotation for unified representations. Our approach trains a teacher and student model connected via distillation. The student GNN encodes protein sequences and structures, while the teacher model leverages GNN and an auxiliary GO encoder to incorporate the functional knowledge, generating hybrid multimodal embeddings passed to the student to learn the function-enriched representations by distribution approximation. Experiments on tasks like protein fold and enzyme commission (EC) prediction show that ProteinSSA significantly outperforms state-of-the-art baselines, demonstrating the benefits of our multimodal framework.

## 1 INTRODUCTION

Proteins are essential molecules that serve as the basic structural and functional components of cells and organisms. A natural protein consists of a linear sequence of amino acids that are linked together by peptide bonds, which folds into a three-dimensional (3D) structure. It is a major scientific challenge to figure out the relationship between a protein's sequence, structure, and function while this knowledge is crucial for elucidating disease mechanisms (Serçinoğlu & Ozbek, 2020). Recent advances like AlphaFold2 (Jumper et al., 2021) have enabled highly accurate protein structure prediction, facilitating the application of artificial intelligence techniques for proteins. As for protein representation learning, it is an active research area that aims to learn underlying patterns from raw protein data for different downstream tasks (Unsal et al., 2022).

Recently, protein language models have been developed to process protein sequences and have demonstrated an ability to learn the certain 'grammar of life' from large numbers of protein sequences (Lin et al., 2022). Models like ProtTrans (Elnaggar et al., 2021) and ESM (Rives et al., 2019; Rao et al., 2021; 2020; Lin et al., 2022) leverage transformers, and attention mechanisms to learn intrinsic patterns in a self-supervised manner, pre-training on large-scale of data. Unlike sequences, protein structures exhibit continuous 3D coordinate data (Fan et al., 2023), requiring different modeling approaches. To represent both 1D sequences and 3D structures, GNN-based models have been designed and adapted (Baldassarre et al., 2021; Hermosilla & Ropinski, 2022). For example, GearNet (Zhang et al., 2023) encodes the sequential and spatial features of proteins by passing messages between nodes and edges in an alternating pattern on multiple types of protein graphs.

Though protein LMs and GNNs have achieved remarkable performance in various protein-related applications, such as tasks of predicting protein stability and EC numbers (Hu et al., 2023). Proteins have more than just sequences and structures. Incorporating functional annotations is also important for enhancing model capabilities and uncovering the intrinsic relationships between protein sequences and functions (Zhou et al., 2023; Hu et al., 2023). Recent works explore token-level protein knowledge from dealing with the functional biomedical texts via protein pre-training (Zhou

et al., 2023; Xu et al., 2023). However, protein sequences vastly outnumber available structures and annotations (Ashburner et al., 2000). For example, there are about 190 thousand structures in the Protein Data Bank (PDB) (Berman et al., 2000b) versus over 500 million sequences in Uni-Parc (Consortium, 2013) and only approximately 5 million GO term triplets in ProteinKG25 (Zhang et al., 2022), including about 600 thousand protein, 50 thousand attribute terms. This scale difference makes it difficult to bring the same success of sequence pre-training into sequence, structure, and function pre-training. In this paper, we utilize the annotation information without relying on pre-training. This allows guiding the sequence-structure model training to learn unified representations for downstream tasks, bypassing the need for immense pre-training.

Considering the data categories and sizes of protein sequences, structures, and GO terms, we propose ProteinSSA, a multimodal framework for protein representation learning. ProteinSSA utilizes a teacher model to learn from sequence-structure-annotation triplets, distilling this knowledge to aid in training the student network. At present, not even 1% of sequenced proteins have functional annotations Torres et al. (2021); Ibtehaz et al. (2023). While the teacher network requires extra functions as input, such information is not always available. The teacher provides functional knowledge, training the sequence-structure student model is more critical as we apply it to downstream tasks for evaluating the framework. To transfer teacher knowledge, we employ domain adaptation techniques to align the embedding distributions between teacher and student. Specifically, we calculate the Kullback-Leibler (KL) divergence to minimize the distance between the distributions of representations from different protein data modalities across the teacher and student domains. The key contributions of this work are threefold:

- We propose ProteinSSA to incorporate multiple types of protein data, including sequence, structure, and functional annotations. This allows learning unified representations without large-scale pre-training, for applicability to various downstream tasks.

- We are the first to adapt the knowledge distillation method to connect the protein teacher-student network, injecting the functional information into the student representations via distribution approximation and domain adaptation.

- We validate ProteinSSA by surpassing current protein representation methods on tasks, including predicting protein fold, enzyme reactions, GO terms, and EC numbers.

## 2 RELATED WORKS

### 2.1 REPRESENTATION LEARNING FOR PROTEIN

Self-supervised pre-training methods have been proposed to learn representations directly from amino acid sequences (Rao et al., 2019)), with significant efforts to increase model or dataset sizes (Rao et al., 2020; Elnaggar et al., 2021; Nijkamp et al., 2022; Ferruz et al., 2022; Rao et al., 2019). To leverage tertiary structures, most works represent sequential and geometric features as the graph node and edge features, using the message passing mechanism to encode them (Zhang et al., 2023; Hermosilla et al., 2021; Jing et al., 2020b). Considering SE(3)-equivariant properties in protein structures, equivariant and invariant features are designed as model inputs (Jing et al., 2020b; Guo et al., 2022a). CDConv (Fan et al., 2023) proposes a continuous-discrete convolution to model the geometry and sequence structures. ProNet (Wang et al., 2023) provides complete geometric representations at multiple tertiary structure levels of granularity. Other works incorporate multi-level structure information (Chen et al., 2023) and multi-task learning (Bepler & Berger, 2019).

Factual biological knowledge has been shown to improve pre-trained language models on protein sequences (Zhang et al., 2022). ProteinBERT (Brandes et al., 2022) are pre-trained on over 100 million proteins and frequent GO annotations from UniRef90 (Boutet et al., 2016). KeAP (Zhou et al., 2023) and ProtST (Xu et al., 2023) train biomedical LMs using masked language modeling (Devlin et al., 2018). Notably, MASSA (Hu et al., 2023) first obtains sequence-structure embeddings from existing pre-trained models (Rao et al., 2020; Jing et al., 2020b), then globally aligns them with GO embeddings using five pre-training objectives. Comparisons are shown in Table 1.

Table 1: Comparisons of existing protein learning methods. A: Annotation, &: and. Note that the input of the student model is without annotations.

| Method | Input Type | Model Type | Pre-training or not |
|---|---|---|---|
| GearNet (Zhang et al., 2023) | Sequence & Structure | GNN | ✔ |
| KeAP (Zhou et al., 2023) | Sequence & A | LM | ✔ |
| MASSA (Hu et al., 2023) | Sequence & Structure & A | LM & GNN | ✔ |
| ProteinSSA (Student) | Sequence & Structure | GNN | ✗ |

## 2.2 KNOWLEDGE DISTILLATION

Knowledge distillation refers to transferring knowledge from a large teacher model to a smaller student model (Hinton et al., 2015). There has been considerable progress in graph-based knowledge distillation, with many proposed methods (Liu et al., 2023; Tian et al., 2022). For instance, RDD (Zhang et al., 2020) forces the student model to directly imitate the full node embeddings of the teacher, transferring more informative knowledge. GraphAKD (He et al., 2022) utilizes adversarial learning to distill node representations from teacher to student, distilling knowledge from both local and global perspectives. It is effective compared to prior graph distillation methods.

## 2.3 DOMAIN ADAPTATION

Domain adaptation generally seeks to learn a model from source-labeled data that can be generalized to a target domain by minimizing differences between domain distributions (Farahani et al., 2021; Wilson & Cook, 2020; Wang & Deng, 2018). Distribution alignment methods minimize marginal and conditional representation distributions between source and target (Nguyen et al., 2022; Long et al., 2015). Adversarial learning approaches have shown impressive performance in reducing divergence between source and target domains (Ganin & Lempitsky, 2015; Long et al., 2018; Pei et al., 2018). Semi-supervised domain adaptation reduces source-target discrepancy given limited labeled target data (Saito et al., 2019; Kim & Kim, 2020; Jiang et al., 2020; Qin et al., 2021). Here, we leverage domain adaptation to align the distributions of representations from teacher and student networks trained on different protein tasks.

## 3 METHODOLOGIES

### 3.1 PRELIMINARIES

In this subsection, we provide the problem definitions and relevant notations. The background knowledge of the local coordinate system is also introduced, which is closely associated with the protein graph edge features.

**Problem Statement**    We represent a protein graph as $G = (\mathcal{V}, \mathcal{E}, X, E)$, where $\mathcal{V} = \{v_i\}_{i=1,...,n}$ and $\mathcal{E} = \{\varepsilon_{ij}\}_{i,j=1,...,n}$ denote the vertex and edge sets with $n$ residues, respectively. We use the coordinate of $C_\alpha$ to represent the position of a residue, and the position matrix is denoted as $\mathcal{P} = \{P_i\}_{i=1,...,n}$, where $P_i \in \mathbb{R}^{3\times 1}$. The node and edge feature matrices are $X = [\boldsymbol{x}_i]_{i=1,...,n}$ and $E = [\boldsymbol{e}_{ij}]_{i,j=1,...,n}$, the feature vectors of node and edge are $\boldsymbol{x}_i \in \mathbb{R}^{d_1}$ and $\boldsymbol{e}_{ij} \in \mathbb{R}^{d_2}$, $d_1$ and $d_2$ are the initial feature dimensions. The GO annotations are denoted as $A = \{A_i\}_{i=1,...,k}$ with $k$ terms in total for proteins, where $A_i \in \{0, 1\}$ is the indicator for annotation $i$. The goal of protein graph representation learning is to form a set of low-dimensional embeddings $z$ for each protein.

There is a source domain $S$ for the teacher model with the data distribution $p_S(z_S|G_S, A)$ in the latent space, and there is also a target domain $T$ for the student model with the data distribution $p_T(z_T|G_T)$ in the latent space. $z_S, z_T$ are latent embeddings from the teacher and student networks for protein graphs $G_S$ and $G_T$.

**Local Coordinate System**    In order to avoid the usage of complicated SE(3)-equivariant models, the invariant and locally informative features are developed from the local coordinate system (Ingra-

ham et al., 2019), shown in Fig 3, which is defined as:

$$\boldsymbol{O}_i = [\boldsymbol{b_i} \quad \boldsymbol{n_i} \quad \boldsymbol{b_i} \times \boldsymbol{n_i}] \tag{1}$$

where $\boldsymbol{u}_i = \frac{P_i - P_{i-1}}{\|P_i - P_{i-1}\|}, \boldsymbol{b_i} = \frac{\boldsymbol{u_i} - \boldsymbol{u_{i+1}}}{\|\boldsymbol{u_i} - \boldsymbol{u_{i+1}}\|}, \boldsymbol{n_i} = \frac{\boldsymbol{u_i} \times \boldsymbol{u_{i+1}}}{\|\boldsymbol{u_i} \times \boldsymbol{u_{i+1}}\|}.$

$$\boldsymbol{e}_{ij} = \text{Concat}(\|P_i - P_j\|, \boldsymbol{O}_i^T \cdot \frac{P_i - P_j}{\|P_i - P_j\|}, \boldsymbol{O}_i^T \cdot \boldsymbol{O}_j) \tag{2}$$

The edge feature vector $\boldsymbol{e}_{ij}$ is the concatenation of the geometric features for protein 3D structures, including distance, direction, and orientation, where $\|\cdot\|$ denotes the $l^2$-norm.

## 3.2 A PRELIMINARY EXPLORATION

For large-scale pre-training, it is unclear whether one or a few self-supervision tasks are sufficient for learning effective representations and which task would be beneficial (Hu et al., 2023). Thus, the performance of pre-trained models is limited by model size, dataset scale, and choice of pre-training tasks. We conducted a preliminary experiment to illustrate this. CDConv (Fan et al., 2023) designs an effective fundamental operation to encapsulate the protein structure without any pre-training or self-supervised learning, achieving comparable accuracy to pre-training methods. It is currently the most effective publicly available method for modeling protein sequence and structure.

In the field of protein pre-training, we select the current state-of-the-art knowledge-enhanced model, KeAP (Zhou et al., 2023), to generate universal sequence-function embeddings, which are used to enhance the CDConv model. ESM-1b (Rives et al., 2019) is the most prevalent sequence pre-training model and is chosen to output sequence embeddings as a comparison with KeAP. By incorporating the embeddings from KeAP and ESM-1b to enhance the embeddings obtained from CDConv, we can compare the quality and performance of the embeddings from these two pre-trained models. The averaged results are shown in Table 2. More details about this experimental settings are provided in Appendix B.1.

Table 2: Accuracy (%) on EC number prediction and GO term prediction. The base model, CD-Conv (Fan et al., 2023), is enhanced by sequence and sequence-function embeddings from ESM-1b (Rives et al., 2019) and KeAP (Zhou et al., 2023).

| Algorithm | GO-BP | GO-MF | GO-CC | EC |
|---|---|---|---|---|
| CDConv | 0.453 | 0.654 | 0.479 | 0.820 |
| Enhanced by the sequence embeddings | 0.471 | 0.665 | 0.538 | 0.862 |
| Enhanced by the sequence-function embeddings | 0.467 | 0.671 | 0.529 | 0.842 |

As shown in Table 2, the sequence embeddings from ESM-1b provide better enhancement compared to the sequence-function embeddings from KeAP when used with CDConv. This observation demonstrates the limitations of the current sequence-function pre-trained model. To overcome these limitations while better utilizing functional information, we propose the multimodal knowledge distillation framework, ProteinSSA.

## 3.3 OVERALL FRAMEWORK

The overall framework of ProteinSSA is illustrated in Figure 1. It consists of two branches that train a teacher model and a student model via iterative knowledge distillation. Compared to the student, the teacher has an additional annotation encoder module comprised of several fully connected layers. This transforms GO annotations into functional embeddings, combined with sequence-structure embeddings from the GNNs to form the final knowledge-enhanced embeddings $z_S$. Previous works have successfully utilized label-augmented techniques to enhance model training (Bengio et al., 2010; Sun et al., 2017). This technique involves encoding labels and combining them with node attributes through concatenation or summation. By doing so, it improves feature representation and enables the model to effectively utilize valuable information from labels.

Instead of directly minimizing distances between sample-dependent embeddings $z_S$ and $z_T$, we develop a sample-independent method. This aligns the student's latent space with the teacher's

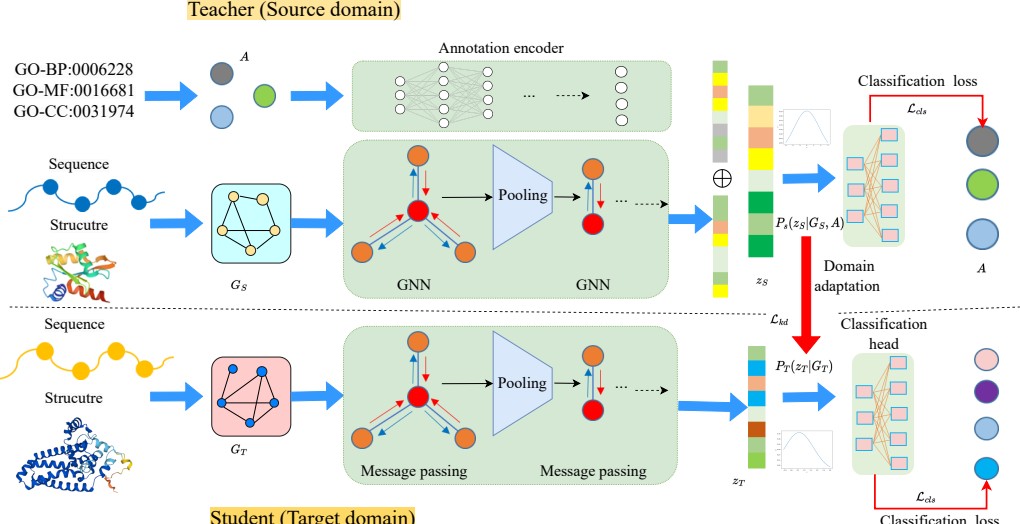

Figure 1: The overall framework of ProteinSSA.

latent space by approximating the distributions of the embeddings obtained from the student and teacher networks. This distribution alignment approach avoids reliance on the input of individual samples. Note that our primary focus is to obtain comprehensive embeddings for the student model, rather than prioritizing the training mode of the teacher model. It can be trained on a larger dataset or multiple datasets, without the need for the student to have access to the same information.

**Protein Graph Message Passing** A protein sequence consists of $n$ residues, which are deemed as graph nodes. We concatenate the one-hot encoding of residue types with the physicochemical properties of each residue, namely, a steric parameter, hydrophobicity, volume, polarizability, isoelectric point, helix probability, and sheet probability (Xu et al., 2022; Hanson et al., 2019), which are used as the graph node features $\boldsymbol{x}_i$. These node features capture meaningful biochemical characteristics, enabling the model to learn which residues tend to be buried, exposed, tightly packed, *etc*. We define the sequential distance, $l_{ij} = \|i - j\|$, and spatial distance $d_{ij} = \|P_i - P_j\|$, where $P_i$ is the 3D coordinate of the $C_\alpha$ atom of the $i$-th residue. There exists an edge between node $v_i$ and $v_j$ if:

$$l_{ij} < l_s \quad \text{and} \quad d_{ij} < r_s \tag{3}$$

where $l_s, r_s$ are predefined radius thresholds, $\boldsymbol{e}_{ij}$ consists of geometric features of the protein structure, defined in Eq. 2. Inspired by CDConv Fan et al. (2023), which convolves node and edge features from sequence and structure simultaneously. We formulate the message passing mechanism as:

$$
\begin{aligned}
\boldsymbol{h}_i^{(0)} &= \text{BN}\left(\text{FC}\left(\boldsymbol{x}_i\right)\right), \\
\boldsymbol{u}_i^{(l)} &= \sigma(\text{BN}(\sum_{v_j \in \mathcal{N}(v_i)} W \boldsymbol{e}_{ij} \boldsymbol{h}_j^{(l-1)})), \\
\boldsymbol{h}_i^{(l)} &= \boldsymbol{h}_i^{(l)} + \text{Dropout}(\text{FC}(\boldsymbol{u}_i^{(l)}))
\end{aligned}
\tag{4}
$$

This mechanism (as shown in Eq. 4) can fuse and update the node and edge features, which include aggregation and update functions, where $\text{FC}(\cdot)$, $\text{BN}(\cdot)$, $\text{Dropout}(\cdot)$ represent fully connected, batch normalization, and dropout layers, $\sigma(\cdot)$ is the activation function LeakyReLU and $W$ is the learnable convolutional kernel. $\mathcal{N}(v_i)$ refers to the neighbors of node $v_i$, and $\boldsymbol{h}_i^{(l)}$ is the representation of node $v_i$ in the $l$-th message passing layer. The node and edge features are processed together in Eq. 4. After message passing operations, a sequence pooling layer is applied to reduce the sequence length, providing a simple but effective way to aggregate key patterns. After average pooling, the residue number is halved; we expand the radius $r_s$ to $2r_s$ to update the edge conditions and perform the message passing and pooling operations again. These operations can make the GNNs cover more distant nodes gradually. The teacher and student models share the same GNNs architecture

to process protein sequences and structures. Finally, a global pooling layer is applied to obtain the graph-level protein embeddings, denoted as $h_S$ and $z_T$ for the teacher and student. Detailed model descriptions are presented in Appendix B.2.

**Protein Domain Adaption**  As shown in Figure 1, the teacher model consists of GNNs, and an auxiliary annotation encoder, which is a multi-layer perceptron (MLP) that provides function-friendly protein representations. The annotations associated with $G_S$ serve as the input for the annotation encoder, resulting in the extraction of feature vector $h_A$. Therefore, we can combine $h_A$ and the graph-level protein embeddings $h_S$ learned from $G_S$ together:

$$h_A = \text{MLP}(A)$$
$$z_S = h_A + \alpha h_S \tag{5}$$

where $\alpha$ is a hyper-parameter, controlling the balance between the contribution of the annotation embeddings $h_A$ and the protein embeddings $h_S$ in the combined representations $z_S$.

As depicted in Figure 1, the generated protein embeddings $z_S$ contain sequence, structure, and function information, guiding the training of the student model. Since knowledge-enhanced embeddings $z_S$ are intended for various protein tasks, they are obtained from the entire protein and GO term datasets. To better capture the inherent uncertainty in the teacher's and student's latent spaces, we calculate distributions within these latent spaces. The minibatch is used to approximate the quantities $p_S(z_S)$ and $p_T(z_T)$:

$$p_S(z_S) = \mathbb{E}_{p_S(G_S, A)}[p(z_S|G_S, A)] \approx \frac{1}{B_S} \sum_{i=1}^{B_S} p_S(z_S|G_S^{(i)}, A^{(i)})$$

$$p_T(z_T) = \mathbb{E}_{p_T(G_T)}[p_T(z_T|G_T)] \approx \frac{1}{B_S} \sum_{i=1}^{B_S} p_T(z_T|G_T^{(i)}) \tag{6}$$

where $B_S$ is the batch size. A Gaussian distribution $\Theta$ is assumed for protein embeddings, which exhibit smoothness and symmetry properties that can reasonably mimic the expected continuity and unimodality of the embeddings aggregated over many residues. We employ the reparameterization trick (Kingma & Welling, 2013) to sample the embeddings.

$$p_S(z_S) = \Theta(\mu_S, \sigma_S^2); \quad p_T(z_T) = \Theta(\mu_T, \sigma_T^2) \tag{7}$$

where $\mu_S, \sigma_S^2$ and $\mu_T, \sigma_T^2$ are the mean and variance values of the embeddings for the teacher and student models, providing a summary of the distribution using first- and second-order statistics.

Proposition 2 in Appendix D shows that the conditional misalignment in the representation space is bounded by the conditional misalignment in the input space. We have:

$$\mathcal{L}_{\text{student}}^* \leq \mathcal{L}_{\text{teacher}} + \frac{M}{\sqrt{2}} \sqrt{\text{KL}\left[p_S(z) \parallel p_T(z)\right] + \mathbb{E}_{p_S(G)}\left[\text{KL}\left[p_S(y|G) \parallel p_T(y|G)\right]\right]} \tag{8}$$

where $\mathcal{L}_{\text{student}}^*$ is the ideal target domain loss, and $\mathcal{L}_{\text{teacher}}$ is the teacher's supervised loss, $M$ is a bound, see Appendix D. $\mathbb{E}_{p_S(G)}\left[\text{KL}\left[p_S(y|G) \parallel p_T(y|G)\right]\right]$ is often small and fixed (not dependent on the representation $z$, and $y$ is the function label). To reduce the generalization bound, we can focus on optimizing the marginal misalignment with a hyper-parameter $\beta$:

$$\mathcal{L}_{\text{teacher}} + \beta(\text{KL}\left[p_S(z) \parallel p_T(z)\right]) \tag{9}$$

Eq. 9 can be used in an unsupervised way for the student to predict functions, which is near the ideal target domain loss. For the proposed framework ProteinSSA (Figure 1), we use the $\mathcal{L}_{\text{teacher}}$ to first train the teacher model, we adopt a hybrid loss $\mathcal{L}$ to train the student model using the labeled data in the target domain, where the $\mathcal{L}_{kd} = \text{KL}\left[p_S(z)|p_T(z)\right]$ is to optimize the marginal misalignment between teacher and student models. Therefore, the final loss $\mathcal{L}$ with a hyper-parameter $\beta$ is formulated as:

$$\mathcal{L} = \mathcal{L}_{\text{student}} + \beta \mathcal{L}_{kd} \tag{10}$$

The objective function of the teacher model $\mathcal{L}_{\text{teacher}}$ is the cross entropy for protein graph classification. It is important to note that the training of the teacher model can be considered distinct from traditional pre-training, as it does not involve unsupervised or self-supervised learning on a large dataset. The hybrid loss of the student model has a cross entropy loss $\mathcal{L}_{\text{student}}$ for classification and a regularization loss $\mathcal{L}_{kd}$ for knowledge distillation.

Table 3: Accuracy (%) of fold classification and enzyme reaction classification. The best results are shown in bold.

| Input | Method | Fold Classification | | | Enzyme |
|---|---|---|---|---|---|
| | | Fold | SuperFamily | Family | Reaction |
| Sequence | CNN (Shanehsazzadeh et al., 2020) | 11.3 | 13.4 | 53.4 | 51.7 |
| | ResNet (Rao et al., 2019) | 10.1 | 7.21 | 23.5 | 24.1 |
| | LSTM (Rao et al., 2019) | 6.41 | 4.33 | 18.1 | 11.0 |
| | Transformer (Rao et al., 2019) | 9.22 | 8.81 | 40.4 | 26.6 |
| Structure | GCN (Kipf & Welling, 2016) | 16.8 | 21.3 | 82.8 | 67.3 |
| | GAT (Velickovic et al., 2017) | 12.4 | 16.5 | 72.7 | 55.6 |
| | 3DCNN_MQA (Derevyanko et al., 2018) | 31.6 | 45.4 | 92.5 | 72.2 |
| Sequence-Structure | GraphQA (Baldassarre et al., 2020) | 23.7 | 32.5 | 84.4 | 60.8 |
| | GVP (Jing et al., 2020a) | 16.0 | 22.5 | 83.8 | 65.5 |
| | ProNet-Amino Acid (Wang et al., 2023) | 51.5 | 69.9 | 99.0 | 86.0 |
| | ProNet-Backbone (Wang et al., 2023) | 52.7 | 70.3 | 99.3 | 86.4 |
| | ProNet-All-Atom (Wang et al., 2023) | 52.1 | 69.0 | 99.0 | 85.6 |
| | GearNet (Zhang et al., 2023) | 28.4 | 42.6 | 95.3 | 79.4 |
| | GearNet-IEConv (Zhang et al., 2023) | 42.3 | 64.1 | 99.1 | 83.7 |
| | GearNet-Edge (Zhang et al., 2023) | 44.0 | 66.7 | 99.1 | 86.6 |
| | GearNet-Edge-IEConv (Zhang et al., 2023) | 48.3 | 70.3 | 99.5 | 85.3 |
| | CDConv (Fan et al., 2023) | 56.7 | 77.7 | 99.6 | 88.5 |
| | ProteinSSA (Student) | **60.5** | **79.4** | **99.8** | **89.4** |

## 4 EXPERIMENTS

### 4.1 TRAINING DETAILS

The proposed multimodal knowledge distillation framework, ProteinSSA, is trained in two steps. We only use about 30 thousand proteins with 2752 GO annotations from the GO dataset, without further division into categories of biological process (BP), molecular function (MF), and cellular component (CC) (Gligorijević et al., 2021). These classes are extracted as input to the teacher model's annotation encoder. we get the $F_{max}$ for the teacher model 0.489 overall. Then, we train the student model. The models are trained with the Adam optimizer using the PyTorch library. Performance is measured with mean values over three initializations. Detailed experimental settings are provided in Appendix B.3.

### 4.2 TASKS AND BASELINES

Following the tasks in IEconv (Hermosilla et al., 2021) and CDConv (Fan et al., 2023), we evaluate ProteinSSA on four protein tasks: protein fold classification, enzyme reaction classification, GO term prediction, and EC number prediction. Detailed task descriptions are presented in Appendix B.4. Dataset statistics are shown in Table 6.

**Baselines** The proposed method is compared with existing protein representation learning methods, which are classified into three categories based on their inputs, which could be a sequence, 3D structure, or both sequence and structure. 1) Sequence-based encoders, including CNN (Shanehsazzadeh et al., 2020), ResNet (Rao et al., 2019), LSTM (Rao et al., 2019) and Transformer (Rao et al., 2019). 2) Structure-based methods (GCN (Kipf & Welling, 2016), GAT (Velickovic et al., 2017), 3DCNN_MQA (Derevyanko et al., 2018) 3) Sequence-structure based models, *e.g.*, GVP (Jing et al., 2020a), ProNet (Wang et al., 2023), GearNet (Zhang et al., 2023), CDConv (Fan et al., 2023), *etc*. GearNet-IEConv and GearNetEdge-IEConv (Zhang et al., 2023) add the IEConv layer on GearNet.

Table 4: $F_{max}$ of GO term prediction and EC number prediction. The best results are shown in bold.

| Category | Method | GO-BP | GO-MF | GO-CC | EC |
|---|---|---|---|---|---|
| Sequence | CNN (Shanehsazzadeh et al., 2020) | 0.244 | 0.354 | 0.287 | 0.545 |
| | ResNet (Rao et al., 2019) | 0.280 | 0.405 | 0.304 | 0.605 |
| | LSTM (Rao et al., 2019) | 0.225 | 0.321 | 0.283 | 0.425 |
| | Transformer (Rao et al., 2019) | 0.264 | 0.211 | 0.405 | 0.238 |
| Structure | GCN (Kipf & Welling, 2016) | 0.252 | 0.195 | 0.329 | 0.320 |
| | GAT (Velickovic et al., 2017) | 0.284 | 0.317 | 0.385 | 0.368 |
| | 3DCNN_MQA (Derevyanko et al., 2018) | 0.240 | 0.147 | 0.305 | 0.077 |
| Sequence-Structure | GraphQA (Baldassarre et al., 2020) | 0.308 | 0.329 | 0.413 | 0.509 |
| | GVP (Jing et al., 2020a) | 0.326 | 0.426 | 0.420 | 0.489 |
| | GearNet (Zhang et al., 2023) | 0.356 | 0.503 | 0.414 | 0.730 |
| | GearNet-IEConv (Zhang et al., 2023) | 0.381 | 0.563 | 0.422 | 0.800 |
| | GearNet-Edge (Zhang et al., 2023) | 0.403 | 0.580 | 0.450 | 0.810 |
| | GearNet-Edge-IEConv (Zhang et al., 2023) | 0.400 | 0.581 | 0.430 | 0.810 |
| | CDConv (Fan et al., 2023) | 0.453 | 0.654 | 0.479 | 0.820 |
| | ProteinSSA (Student) | **0.464** | **0.667** | **0.492** | **0.857** |

## 4.3 RESULTS OF FOLD AND ENZYME REACTION CLASSIFICATION.

Table 3 shows performance comparisons on protein fold and enzyme reaction prediction across different methods, reported as average values. From the table 3, we can see that the proposed ProteinSSA achieves the best performance among all methods on the four test sets for both fold and reaction prediction tasks. Sequence-structure based methods generally outperform sequence- or structure-only methods, indicating the benefits of co-modeling sequence and structure. Notably, on the Fold test set, ProteinSSA improves accuracy by over 6.7% compared to prior state-of-the-art techniques, demonstrating its effectiveness at learning sequence, structure and function mappings. Additionally, CDConv ranks second among the methods, with both it and ProteinSSA using sequence-structure convolution architectures. This suggests the teacher-student training paradigm in ProteinSSA helps the student learn superior protein embeddings.

## 4.4 RESULTS OF GO TERM AND EC NUMBER PREDICTION

Following the protocol in GearNet (Zhang et al., 2023), the test sets for GO term and EC number prediction only contain PDB chains with less than 95% sequence identity to the training set, ensuring rigorous evaluation. The student model conducts the experiments, and the teacher model's annotations are not classified into these classes, avoiding data leakage. Table 4 shows comparative results between different protein modeling methods on these tasks, with performance measured by $F_{max}$, which balances precision and recall, working well even if positive and negative classes are imbalanced. The mean values of three independent runs are reported. ProteinSSA achieves the highest $F_{max}$ across all test sets for both GO and EC prediction, outperforming state-of-the-art approaches. This demonstrates ProteinSSA's strong capabilities for predicting protein functions and activities. Compared to preliminary results in Table 2, ProteinSSA even exceeds CDConv (Fan et al., 2023) augmented with sequence-function embeddings from the large-scale pre-trained model, KeAP (Zhou et al., 2023) on EC number prediction, while being comparable on GO term prediction. Overall, the consistent improvements verify the benefits of injecting function information into sequence-structure models, as done in ProteinSSA's teacher-student framework. The results cement ProteinSSA's effectiveness using knowledge distillation techniques.

## 4.5 ABLATION STUDY

Table 5 presents ablation studies of the proposed ProteinSSA model on the four downstream tasks. We examine the impact of removing the teacher model, which means removing the $\mathcal{L}_{kd}$. We also remove the annotation encoder in the teacher, which means that we incorporate function information into the loss function for the teacher models. As shown in Table 5, removing the teacher model

Table 5: Ablation experiments of our proposed method. w/o AE-T denotes without the annotation encoder in the teacher model. w/o teacher means without the teacher model and directly using the student model, which means without $\mathcal{L}_{kd}$.

| Method | Fold Classification | | | Enzyme | GO | | | EC |
|---|---|---|---|---|---|---|---|---|
| | Fold | Superfamily | Family | Reaction | BP | MF | CC | |
| ProteinSSA | 60.5 | 79.4 | 99.8 | 89.4 | 0.464 | 0.667 | 0.492 | 0.857 |
| w/o AE-T | 60.4 | 79.1 | 99.7 | 88.9 | 0.454 | 0.664 | 0.490 | 0.854 |
| w/o Teacher | 57.8 | 78.7 | 99.6 | 88.6 | 0.458 | 0.660 | 0.484 | 0.851 |

altogether (w/o Teacher) leads to substantial performance drops across all tasks compared to the full ProteinSSA. This shows the teacher's knowledge distillation provides useful signals for the student model. Besides, removing the annotation encoder in the teacher (w/o AE-T) also degrades performance, though less severely. This indicates the annotation encoder slightly helps align teacher outputs with the downstream tasks. These ablations highlight the importance of utilizing the teacher model and the annotation encoder for optimal results.

Figure 2 shows the comparisons of the knowledge distillation loss $\mathcal{L}_{kd}$, with and without being involved in backpropagation during training. When the loss $\mathcal{L}_{kd}$ is not involved in the process of the gradient backpropagation, it decreases due to the decreasing classification loss $\mathcal{L}_{\text{student}}$, but remains much higher than when $\mathcal{L}_{kd}$ is involved. This validates the effectiveness of the proposed knowledge distillation loss and its role in training.

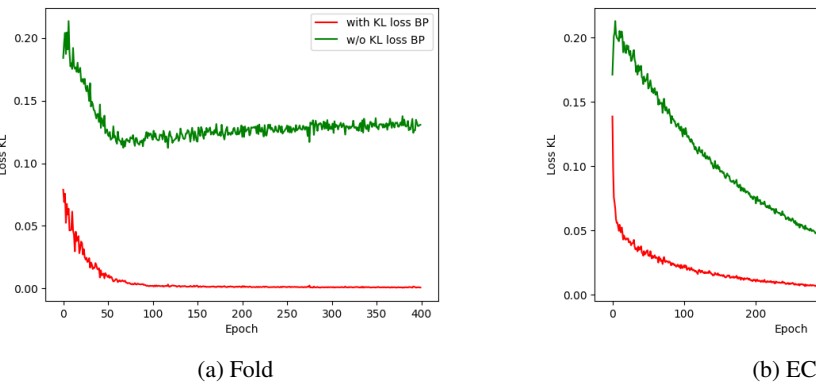

(a) Fold                                             (b) EC

Figure 2: The KL training loss curves on the fold classification and EC number prediction. The red curve denotes $\mathcal{L}_{kd}$ conducts its function, while the blue curve denotes we calculated the value of $\mathcal{L}_{kd}$, but it is not involved in the process of the gradient backpropagation (BP).

## 5 CONCLUSION

In this paper, we propose ProteinSSA, a multimodal protein representation learning framework integrating the information from protein sequences, structures, and annotations. Importantly, we estimate the latent embedding distributions for the teacher-student model and learn annotation-enriched student representations by distribution approximation. Compared to mainstream protein representation learning techniques, ProteinSSA achieves superior performance in predicting protein structure, reactions, GO terms, and EC numbers. The consistent improvements across benchmarks highlight the advantages of this approach for informative protein representation learning. However, ProteinSSA uses predefined and fixed weight parameters, which need empirical tuning and experimental validations. Additionally, the student is restricted by the teacher's ability. Therefore, this framework could be improved by training the teacher on larger annotation datasets.

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

## A  LOCAL COORDINATE SYSTEM

We have introduced the local coordinate system (Ingraham et al., 2019) $Q_i$ in the Section 3.1, which defines the geometric properties of the point $v_i$. It is shown in Figure 3. From this figure, we can easily find that $b_i$ is the negative bisector of the angle between the rays $(P_{i-1} - P_i)$ and $(P_{i+1} - P_i)$.

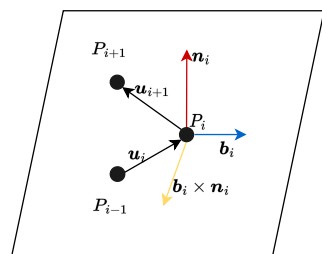

## B  EXPERIMENT SETUP

### B.1  DETAILED EXPERIMENT SETTING IN SECTION 3.2

As stated in Section 3.2, embeddings generated from the pre-trained models, ESM-1b[1] (Rives et al., 2019) and KeAP[2] (Zhou et al., 2023), are used to enhance the sequence-structure model CD-Conv (Fan et al., 2023). As shown in Figure 4. A two-layer MLP is used to encode the generated embeddings, which are then added to the CDConv embeddings. The MLP has feature dimensions of 1024 and 2048, with other hyper-parameters remaining the same as the base models. This allows the integration of knowledge from large-scale pre-trained language and protein models into the sequence-structure framework for improved protein characterization.

Figure 3: The local coordinate system $Q_i$ related to protein graph node $v_i$, $P_i$ is the coordinate of residue $i$.

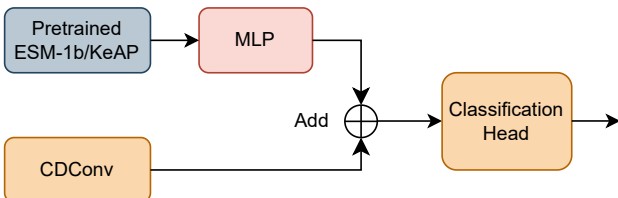

Figure 4: An illustration of the enhanced CDConv model.

### B.2  MODEL DETAILS

The radius $r_s$ threshold increases from 4 to 16, and $l_s$ is set to 11. We set two message passing layers with one average sequence pooling per GNN. After the pooling layer, the number of residues is halved, and we update the edge conditions before performing another round of message passing and pooling, as illustrated in Figure 1. The final GNNs include eight message-massing and four pooling layers, which are sufficient for achieving satisfactory results. The number of initial feature channels is 256, increased to 2048. The annotation encoder has 2 FC layers changing feature channels from 2752 to 2048. The classification head is a liner layer for predicted classes. For the teacher model, we use $z_S$ to get the predicted annotations by the classification head and calculate the loss by $\mathcal{L}_{\text{teacher}}$. The final loss $\mathcal{L}$ is used for the training of the student model.

As shown in Figrue 5, residues that are spatially adjacent can still exist even when the sequence distance is large. The medians suggest the sequence-structure distance may have a linear relationship. Thus, we perform sequence average pooling, and change edge conditions after once pooling. These operations enable the protein graph to cover more distant nodes.

### B.3  TRAINING DETAILS

Dataset statistics (Zhang et al., 2023) of the four downstream tasks are summarized in Table 6. The proposed framework conducted experiments on NVIDIA-SMI A100 GPUs and NVIDIA Tesla V100 GPUs, implemented with PyTorch 1.13+cu117 and PyTorch Geometric 2.3.1 with CUDA 11.2.

---

[1]https://github.com/facebookresearch/esm#available-models
[2]https://github.com/RL4M/KeAP

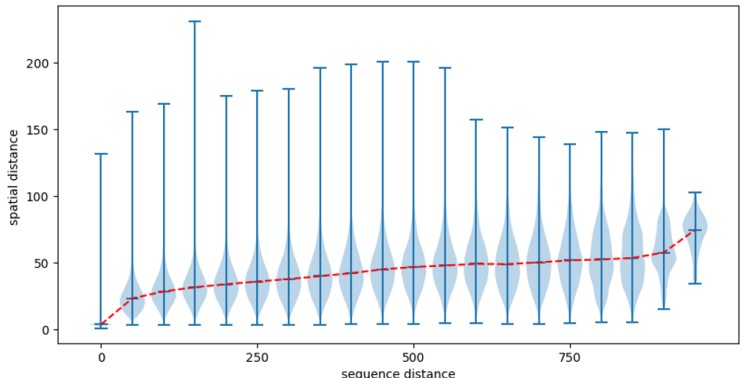

Figure 5: The relationships of distances between sequence and structure on the EC dataset, the sequential distance $l_{ij}$ is from 1 to $n$-1, and the x-axis means $l_{ij} - 1$, the y-axis means $d_{ij}$. The dashed red line connects their median values.

Table 6: Dataset statistics. #X means the number of X.

| Dataset | #Train | #Validation | #Test |
|---|---|---|---|
| Enzyme Commission | $15,550$ | $1,729$ | $1,919$ |
| Gene Ontology | $29,898$ | $3,322$ | $3,415$ |
| Fold Classification - Fold | $12,312$ | $736$ | $718$ |
| Fold Classification - Superfamily | $12,312$ | $736$ | $1,254$ |
| Fold Classification - Family | $12,312$ | $736$ | $1,272$ |
| Reaction Classification | $29,215$ | $2,562$ | $5,651$ |

In biology, a linear combination of original data with Gaussian noise (Guo et al., 2022b) is a simple but effective way to augment the protein data:

$$(P_i, \boldsymbol{x}_i) \leftarrow (P_i, \boldsymbol{x}_i) + \Theta, \Theta \sim (\mu_k, \sigma_k^2) \tag{11}$$

where $\mu_k$ and $\sigma_k$ are selected as the random noise's mean (expectation) and standard deviation.

Hyper-parameters related to the networks are set the same across different datasets: Adam optimizer with learning rate $l_r = 1e - 3$, weight decay $decay = 5e - 4$, epochs $T = 300$, Gaussian noise $\mu_k = 0, \sigma_k = 0.1$, it indicates trivial perturbation is introduced to the protein native structures.

The other dataset-specific hyper-parameters are determined by an AutoML toolkit NNI (Microsoft, 2021) with the search spaces. The loss weight hyper-parameter is related to the value of the task-specific loss $\beta = \{1, 0.1, 0.01, 0.001, 0\}$, and $\alpha = \{10, 1, 0.1, 0.01, 0.001, 0\}$. As for the batch size and training epochs, *etc.*, which influence the convergence speed of deep learning models, details about implementation on the NVIDIA-SMI A100 GPUs are shown in Table 7.

Table 7: More details of training setup

| Hyper-parameter | Fold | Enzyme Reaction | GO | EC |
|---|---|---|---|---|
| Batch size | 16 | 8 | 24 | 64 |
| Epoch | 400 | 400 | 500 | 500 |

### B.4 TASK INTRODUCTION

**Fold Classification** In order to understand how protein structure and evolution interact, it is crucial to be able to predict fold classes (Hou et al., 2018). This dataset contains 16,712 total proteins across 1,195 fold classes. Three test sets are provided. Fold: proteins from the same superfamily are excluded during training; SuperFamily: proteins from the same family are not used for training; and Family: the training set includes proteins from the same family.

**Enzyme Reaction Classification**  Enzyme reaction classification can be viewed as a protein function prediction task based on the enzyme-catalyzed reactions defined by the four levels of enzyme commission numbers (Webb et al., 1992; Omelchenko et al., 2010). We use the dataset (Hermosilla et al., 2021; Berman et al., 2000a) containing 29,215 training proteins, 2,562 validation proteins, and 5,651 test proteins, spanning 384 four-level EC classes.

**GO Term Prediction**  The aim of GO term prediction is to predict whether a given protein should be annotated with a particular GO term. As we have stated before, proteins are categorized into three hierarchical ontologies: MF, BP and CC. Specifically, MF denotes molecular activities of a protein, BP refers to larger biological processes it is involved in, and CC describes subcellular locations and extracellular components (Bateman, 2019). Accurately assigning GO terms is crucial for understanding protein function and assessing computational methods.

**EC Number Prediction**  This task aims to predict the 538 EC numbers at the third and fourth level hierarchies for different proteins (Gligorijević et al., 2021), which provide precise information about a protein's enzymatic function, based on the protein's features. The large number of classes at the third and fourth EC levels makes this a challenging multi-class prediction problem in bioinformatics.

## C  EVALUATION METRIC

$F_{\max}$ provides an overall metric that combines both accuracy and coverage of the predictions. It is calculated by first determining the precision and recall for each protein, then averaging these results over all proteins (Zhang et al., 2023; Gligorijević et al., 2021). $p_i^j$ is the prediction probability for the $j$-th class of the $i$-th protein, given the decision threshold $t \in [0, 1]$, the precision and call are given as:

$$\text{precision}_i(t) = \frac{\sum_j \mathbb{I}[((p_i^j \geq t) \cap b_i^j)]}{\sum_j \mathbb{I}[(p_i^j \geq t)]}, \quad \text{recall}_i(t) = \frac{\sum_j \mathbb{I}[((p_i^j \geq t) \cap b_i^j)]}{\sum_j b_i^j}$$

where $b_i^j \in \{0, 1\}$ is the corresponding binary class label, and $\mathbb{I} \in \{0, 1\}$ is an indicator function. If there are $N$ proteins in total, these protein-level precision and recall values are averaged over all proteins to obtain the overall precision and recall for the dataset, then the average precision and recall are defined as:

$$\text{precision}(t) = \frac{\sum_i^N \text{precision}_i(t)}{\sum_i^N \left( \left( \sum_j \left( p_i^j \geq t \right) \right) \geq 1 \right)}, \quad \text{recall}(t) = \frac{\sum_i^N \text{recall}_i(t)}{N}$$

Finally, $F_{\max}$ is defined as the maximum value of F-score over all thresholds.

$$F_{\max} = \max_t \left\{ \frac{2 \cdot \text{precision}(t) \cdot \text{recall}(t)}{\text{precision}(t) + \text{recall}(t)} \right\} \tag{12}$$

## D  KL GUIDED DOMAIN ADAPTATION

Assuming source and target domains have the same support set and share the representation mapping $p(z|G)$, this means these two domains have the same datasets of protein graphs and functions. Given the representation $z$, we learn a classifier to predict the label $y$ through the predictive distribution $\hat{p}(y|z)$ that is an approximation of the ground truth. During training, the representation network $p(z|G)$ and the classifier $\hat{p}(y|z)$ are trained jointly on the source domain and we hope that they can generalize to the target domain, meaning that both $p(z|G)$ and $\hat{p}(y|z)$ are kept unchanged between training and testing.

We define the predictive distribution of $y$ given $G$ as

$$\hat{p}(y|G) = \mathbb{E}_{p(z|G)}[\hat{p}(y|z)] \tag{13}$$

We have a single $z$ from the source model $p(z|G)$ for each protein. The training objective of the source domain is

$$\mathcal{L}_{\text{teacher}} = \mathbb{E}_{G, y \sim p_S(G,y), z \sim p(z|G)}[-\log \hat{p}(y|z)] = \mathbb{E}_{p_S(z,y)}[-\log \hat{p}(y|z)] \tag{14}$$

We consider the two assumptions of the representation $z$ on the source domain:

**Assumption 1.** $I_S(z, y) = I_S(G, y)$, where $I_S(\cdot, \cdot)$ is the mutual information term, calculated on the source domain. In particular:

$$I_S(z, y) = \mathbb{E}_{p_S(z,y)} \left[ \log \frac{p_S(z, y)}{p_S(z) p_S(y)} \right]; \quad I_S(G, y) = \mathbb{E}_{p_S(G,y)} \left[ \log \frac{p_S(G, y)}{p_S(G) p_S(y)} \right] \quad (15)$$

The mutual information quantifies the amount of information shared between the variables $z$ and $y$ (or $G$ and $y$) in the source domain. It measures the dependence or correlation between these variables in the context of the source domain data. This is often referred to as the 'sufficiency assumption' since it indicates that the representation $z$ has the same information about the label $y$ as the original input protein graph $G$, and is sufficient for this prediction task in the source domain. Note that the data processing inequality indicates that $I_S(z, y) \leq I_S(G, y)$, so here we assume that $z$ contains maximum information about $y$.

**Assumption 2.** $p_S(y|G) = \mathbb{E}_{p(z|G)} [p_S(y|z)]$

When this assumption holds, the predictive distribution $\hat{p}(y|G)$ will approximate $p_S(y|G)$, as long as $\hat{p}(y|z)$ approximates $p_S(y|z)$.

The above two assumptions ensure that the teacher network has good performance in the source domain. Now, we continue to consider the test loss and how we can reduce it. The loss of the target domain is:

$$\begin{aligned} \mathcal{L}_{\text{student}}^* &= \mathbb{E}_{p_T(G,y)}[-\log \hat{p}(y|G)] = \mathbb{E}_{p_T(G,y)} \left[ -\log \mathbb{E}_{p(z|G)}[\hat{p}(y|z)] \right] \\ &\leq \mathbb{E}_{p_T(G,y)} \left[ \mathbb{E}_{p(z|G)}[-\log \hat{p}(y|z)] \right] \\ &= \mathbb{E}_{p_T(z,y)}[-\log \hat{p}(y|z)] \end{aligned} \quad (16)$$

Since we do not know the target domain and the target data distribution, there is no way to guarantee the invariance (both marginally and conditionally) of the representation $z$. Therefore, We introduce the following proposition that ensures a generalization bound of the target domain loss based on the source domain loss and the KL divergence:

**Proposition 1.** If the loss $-\log \hat{p}(y|z)$ is bounded by $M$, we have:

$$\begin{aligned} \mathcal{L}_{\text{student}}^* &\leq \mathcal{L}_{\text{teacher}} + \frac{M}{\sqrt{2}} \sqrt{\text{KL}\left[p_S(y, z) \parallel p_T(y, z)\right]} \\ &= \mathcal{L}_{\text{teacher}} + \frac{M}{\sqrt{2}} \sqrt{\text{KL}\left[p_S(z) \parallel p_T(z)\right] + \mathbb{E}_{p_S(z)}\left[\text{KL}\left[p_S(y|z) \parallel p_T(y|z)\right]\right]} \end{aligned} \quad (17)$$

**Proposition 2.** If Assumption 1 and 2 hold, and if $\frac{p_S(G,y)}{p_T(G,y)} < \infty$ (i.e., there exists $N$, which can be arbitrarily large, such that $\frac{p_S(G,y)}{p_T(G,y)} < N$), we have

$$\mathbb{E}_{p_S(G)}\left[\text{KL}\left[p_S(y|z) \parallel p_T(y|z)\right]\right] \leq \mathbb{E}_{p_S(G)}\left[\text{KL}\left[p_S(y|G) \parallel p_T(y|G)\right]\right] \quad (18)$$

This shows that the conditional misalignment in the representation space is bounded by the conditional misalignment in the input space. It then follows that:

$$\mathcal{L}_{\text{student}}^* \leq \mathcal{L}_{\text{teacher}} + \frac{M}{\sqrt{2}} \sqrt{\text{KL}\left[p_S(z) \parallel p_T(z)\right] + \mathbb{E}_{p_S(G)}\left[\text{KL}\left[p_S(y|G) \parallel p_T(y|G)\right]\right]} \quad (19)$$

We know $y$ can represent the underlying functional label for the student model. Although the student model may not have these functional labels, but we can assume that they exist for theoretical reasons. The derived misalignment Eq. 19 and the derived loss Eq. 8 are based on the assumption that the source and target domains have the same support set. Thus, the loss of Eq. 8 can be used in an unsupervised way for the student to predict functions. However, the student model is applied to different downstream tasks, like classification, which has classification classes. Thus, we add the supervised student loss $\mathcal{L}_{\text{student}}$ and the knowledge distillation loss the $\mathcal{L}_{kd}$ as the final hybrid loss for the student to improve its performance on classification tasks.

# E  TEACHER MODEL RESULTS

As we have mentioned in Section 4.1, we train the teacher model using approximately 30K proteins from the GO dataset without separating the annotations into BP, MF, and CC. Overall, we achieved a $F_{max}$ score of 0.489 for the teacher model. As shown in Figure 1, the inputs of the teacher model consist of sequence, structure, and function. However, we currently lack functional data for tasks such as fold classification, enzyme reaction classification, and EC number prediction. Therefore, we evaluate the teacher model using the GO dataset and calculate scores for BP, MF, and CC. The results are shown in Figure 6.

The goal of the GO term prediction task is to identify protein functions. MF has 1,943 classes. BP is categorized into 489 classes. CC is classified into 320 classes. The difference between the teacher and the student is that there is an additional annotation encoder in the teacher model. From the provided Figure 6, it is evident that incorporating functional information as the input of the annotation encoder significantly enhances performance, particularly for MF and CC. These two classes have fewer categories and are more accessible, resulting in higher scores. These results demonstrate the effectiveness of the teacher model and the label-augmented technique, which can encode the functions into embeddings to improve protein feature representations.

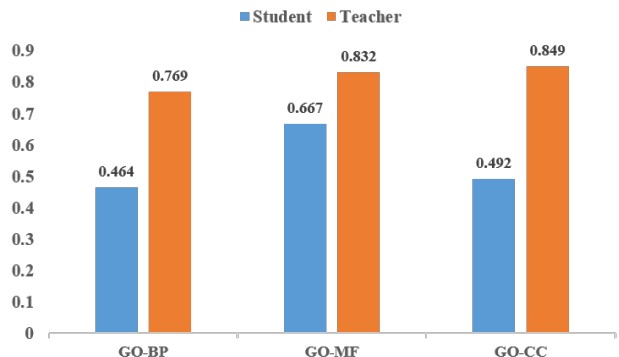

Figure 6: Comparisons of the teacher and student of ProteinSSA on GO term prediction.

Table 8: Accuracy (%) of fold classification and enzyme reaction classification. The best results are shown in bold. Param., means the number of trainable parameters (B: billion; M: million; K: thousand).

| Method | Param. | Pre-training Dataset (Size) | Fold Classification | | | Enzyme |
|---|---|---|---|---|---|---|
| | | | Fold | SuperFamily | Family | Reaction |
| DeepFRI | 6.2M | Pfam (10M) | 15.3 | 20.6 | 73.2 | 63.3 |
| ESM-1b | 650M | UniRef50 (24M) | 26.8 | 60.1 | 97.8 | 83.1 |
| ProtBERT-BFD | 420M | BFD (2.1B) | 26.6 | 55.8 | 97.6 | 72.2 |
| IEConv (amino level) | 36.6M | PDB (476K) | 50.3 | **80.6** | 99.7 | 88.1 |
| GearNet (Multiview Contras) | 42M | AlphaFoldDB (805K) | 54.1 | 80.5 | **99.9** | 87.5 |
| GearNet (Residue Type) | 42M | AlphaFoldDB (805K) | 48.8 | 71.0 | 99.4 | 86.6 |
| GearNet (Distance) | 42M | AlphaFoldDB (805K) | 50.9 | 73.5 | 99.4 | 87.5 |
| GearNet (Angle) | 42M | AlphaFoldDB (805K) | 56.5 | 76.3 | 99.6 | 86.8 |
| GearNet (Dihedral) | 42M | AlphaFoldDB (805K) | 51.8 | 77.8 | 99.6 | 87.0 |
| ProteinSSA | 100M | - | **60.5** | 79.4 | 99.8 | **89.4** |

# F  COMPARISON WITH PRETRAINING METHODS

In the teacher-student framework, the teacher model is usually a well-learned model that serves as a source of knowledge for the student model. The student model aims to mimic the behavior or predictions of the teacher model. ProteinSSA uses annotations for the teacher model, its objective is to learn embeddings in the latent space that contain functional information and provide intermediate supervision during knowledge distillation for the student model. Therefore, the complete training

Table 9: $F_{max}$ of GO term prediction and EC number prediction. The best results are shown in bold. Param., means the number of trainable parameters (B: billion; M: million; K: thousand).

| Method | Param. | Pre-training Dataset (Size) | GO BP | MF | CC | EC |
|--------|--------|------------------------------|-------|-----|-----|-----|
| DeepFRI | 6.2M | Pfam (10M) | 0.399 | 0.465 | 0.460 | 0.631 |
| ESM-1b | 650M | UniRef50 (24M) | 0.470 | 0.657 | 0.488 | 0.864 |
| ProtBERT-BFD | 420M | BFD (2.1B) | 0.279 | 0.456 | 0.408 | 0.838 |
| LM-GVP | 420M | UniRef100 (216M) | 0.417 | 0.545 | **0.527** | 0.664 |
| IEConv (amino level) | 36.6M | PDB (476K) | 0.468 | 0.661 | 0.516 | - |
| GearNet (Multiview Contras) | 42M | AlphaFoldDB (805K) | **0.490** | 0.654 | 0.488 | 0.874 |
| GearNet (Residue Type) | 42M | AlphaFoldDB (805K) | 0.430 | 0.604 | 0.465 | 0.843 |
| GearNet (Distance) | 42M | AlphaFoldDB (805K) | 0.448 | 0.616 | 0.464 | 0.839 |
| GearNet (Angle) | 42M | AlphaFoldDB (805K) | 0.458 | 0.625 | 0.473 | 0.853 |
| GearNet (Dihedral) | 42M | AlphaFoldDB (805K) | 0.458 | 0.626 | 0.465 | 0.859 |
| KeAP | 420M | ProteinKG25 (5M) | 0.466 | 0.659 | 0.470 | 0.845 |
| ESM-2 | 650M | UniRef50 (24M) | 0.472 | 0.662 | 0.472 | 0.874 |
| ProtST-ESM-1b | 759M | ProtDescribe (553K) | 0.480 | 0.661 | 0.488 | 0.878 |
| ProtST-ESM-2 | 759M | ProtDescribe (553K) | 0.482 | **0.668** | 0.487 | **0.878** |
| ProteinSSA | 100M | - | 0.464 | 0.667 | 0.492 | 0.857 |

of the teacher model is not our primary concern. Our main focus is to obtain comprehensive embeddings for the student model, which is trained using distillation loss and task loss without the annotations input. As we have mentioned earlier, the training of the teacher model can still be seen as training instead of pre-training because it does not involve unsupervised or self-supervised learning on a large dataset. As discussed in Section 3.2, we highlight the limitations of pre-training and the absence of a well-learned protein functional encoder to encode functional information. Only a few sequenced proteins have functional annotation. While the teacher network requires extra functions as input, such information is not always available. To address these challenges and make better use of functional information without extensive pre-training, we propose ProteinSSA.

To show its effectiveness, we compare the proposed ProteinSSA (student) to pre-training or self-supervised learning methods: DeepFRI Gligorijević et al. (2021), ESM-1b Rives et al. (2019), ProtBERT-BFD Elnaggar et al. (2021), LM-GVP Wang et al. (2021), amino-acid level IEConv Hermosilla & Ropinski (2022), GearNet (GearNet-Edge-IEConv) Zhang et al. (2023), ESM-2 Lin et al. (2022), KeAP Zhou et al. (2023), and ProtST Xu et al. (2023) on these four tasks, including protein fold classification, enzyme reaction classification, GO term prediction, and EC number prediction.

The results are shown in Table 8 and Table 9. Without any pre-training or self-supervised learning, our proposed framework, ProteinSSA, achieves comparable accuracy with those methods with less trainable parameters and even outperforms them on the fold and enzyme reaction classification.

