# OpenReview forum: "Multimodal Distillation of Protein Sequence, Structure, and Function"
_ICLR.cc/2024/Conference — Submitted to ICLR 2024_

### Official Review · Reviewer_G223 · 2023-10-29

**Soundness:** 3 good
**Presentation:** 3 good
**Contribution:** 3 good
**Rating:** 6
**Confidence:** 4

**Summary:**

The paper proposes a new method for multimodal training of protein models based on a distillation method. The multimodal model incorporates Protein Sequence, Structure, Gene Ontology Annotation - named Protein SSA for short, which is tested on protein fold and enzyme commission tasks.

The paper first introduces the problem settings of modeling protein properties and behavior using machine learning with an additional focus on how multimodal data sources can enhance the modeling performance. This leads to the paper's key claims that prior work did not incorporate all possible modalities into their methods prompting the creation of Protein SSA which includes a broader set of modalities. The authors also claim that there is a shortage of protein modeling methods that do not require costly pretraining, leading them to propose a knowledge distillation based training for their multimodal setting. The paper then discusses related work in protein representation learning, domain adaptation and knowledge distillation method with a particular focus on graph-based knowledge distillation methods given that the paper trains GNN in their method.

Next, the paper describes the problem setting and provides a preliminary exploration on whether multimodal embeddings can enhance performance on relevant protein tasks (GO, EC) with the evidence generally being supportive. The paper then describes the main method contribution in Protein SSA, including relevant formulation of message passing for the protein graph as well as the domain adaptation and knowledge distillation framework. The knowledge distillation framework mainly relies on minimizing the KL divergence between the embeddings of the teacher and student models, both of which are approximated by Gaussian distributions.

In Section 4, the paper describes the experiments Fold classification and enzyme reaction, as well as on GO and EC prediction tasks. Compared to the baselines presented in the paper, Protein SSA generally performs best across all tasks, including different types of methods that use a lower number of modalities. Section 4.5 of the experiments includes an ablation study where the paper investigates the importance of different components, including the presence of annotation in the teacher model, the presence of the teacher model itself and training without the KL loss.

**Strengths:**

The paper has the following strengths:
* It proposes a new method (ProteinSSA) for multimodal protein modeling that includes a larger set of modalities that taken together improve modeling performance (originality, significance).
* The problem definition and relevant related work are extensively discussed (quality, clarity).
* The paper includes a relevant ablation study that investigates the effect of removing different components of ProteinSSA (quality, significance).
* The experimental results are generally nicely presented with relevant analysis provided (quality, significance)

**Weaknesses:**

The main weakness of the paper is clarify surrounding the training method used:
* It is unclear whether ProteinSSA makes use of pretrained embedding model, especially for the teacher model. The paper mentions training ProteinBERT with additional modalities, but generally claims that ProteinSSA does not require large-scale pretraining. This appears inconsistent and further clarification would be helpful (significance, clarity).
* The paper does not compare results against larger scale protein models for relevant tasks, including the ones mentioned in related work (e.g., ESM, KeAP, ProtST). It would be good to get a sense of much model size affects performance on the studied tasks (quality, significance).
* The GNN architecture is not fully described (clarity).

**Questions:**

* Can you describe how you obtain the embeddings for each modality? Do you use pretrained models for some or all modalities?
* Can you describe how large your model ends up being in terms number of trainable parameters?
* Can you describe your GNN architecture in more detail? How do you consolidate the graphs from the different graph modalities (sequence, structure) into joint embeddings?
* Can you add the performance of the teacher model into your results tables?

---

> ### Author Response · Authors · 2023-11-20
>
> Dear Reviewer G223,
>
> We are grateful for your thorough review. Your comments are highly valued, and we would like to express our heartfelt gratitude. We do our utmost to address the questions you have raised:
>
> **Q1** It is unclear whether ProteinSSA makes use of pretrained embedding model, especially for the teacher model. The paper mentions training ProteinBERT with additional modalities, but generally claims that ProteinSSA does not require large-scale pretraining. Can you describe how you obtain the embeddings for each modality? Do you use pretrained models for some or all modalities?
>
> **A1** Thank you for your valuable feedback! (1) ProteinBERT [1] is not a part of ProteinSSA, and ProteinSSA does not make use of pre-trained embedding model for both the teacher model and the student model. (2) ProteinBERT is the first model to utilize the protein annotation information, using 8943 frequent Gene Ontology (GO) annotations. Compared with ProteinBERT, we use 2752 GO annotations from the GO dataset. We have modified this in the revised manuscript to decrease the misleading. The current state-of-the-art (SOTA) sequence-function pre-trained model is the KeAP [2], but the SOTA model KeAP fails to generate protein embeddings that surpass those obtained from the sequence pre-trained model ESM-1b [10]. This observation demonstrates the limitations of the sequence-function pre-trained model and motivates this work. (3) The main objective of the teacher model is to learn embeddings that contain functional information and provide intermediate supervision during knowledge distillation for the student model. Therefore, the complete training of the teacher model is not our primary concern. (4) Pre-training involves learning general representations from a large dataset. In this work, the teacher model is trained on a relatively small dataset consisting of about 30 thousand proteins with 2752 GO annotations. This is in comparison to ProteinBERT, which uses 106 million proteins, and KeAP, which uses 5 million triplets. (5) There are three protein modalities that we used, including sequence, structure, and function. Both the student and teacher networks use the proposed protein graph message-passing and pooling layers to process the sequence and structure. Thus, the sequence-structure embeddings are obtained from the designed graph neural network. In the teacher model, the function information is encoded by the annotation encoder to get the functional embeddings.
>
>
> **Q2** The paper does not compare results against larger scale protein models for relevant tasks, including the ones mentioned in related work (e.g., ESM, KeAP, ProtST). It would be good to get a sense of much model size affects performance on the studied tasks. Can you describe how large your model ends up being in terms number of trainable parameters?
>
> **A2** Thank you for your informative reviews! (1) Different from those pretraining or self-supervised learning works, our method is to obtain comprehensive embeddings for the student model, which have the functional information learned by the distillation loss. To show the effectiveness, we compare the proposed ProteinSSA to pretraining or self-supervised learning methods: DeepFRI [4], ESM-1b [10], ProtBERT-BFD [5], LM-GVP [6], amino-acid-level IEConv [7], GearNet-based methods [8], ESM-2 [3], KeAP [2], and ProtST [9] on these four tasks, including protein fold classification, enzyme reaction classification, Gene Ontology (GO) term prediction, and enzyme commission (EC) number prediction. Param. means the number of trainable parameters (B: billion; M: million; K: thousand).
>
> | Method | Param.| Pretraining Dataset |Fold | SuperFamily | Family | Enzyme Reaction |
> | :--- | :---: | :---: | :---: | :---: | :---: |:---: |
> |DeepFRI | 6.2M| Pfam (10M)| 15.3 | 20.6 | 73.2 | 63.3 |
> |ESM-1b | 650M| UniRef50 (24M)| 26.8 | 60.1 | 97.8 | 83.1 |
> |ProtBERT-BFD| 420M| BFD (2.1B)| 26.6 | 55.8 | 97.6 | 72.2 |
> | IEConv (amino level)| 36.6M | PDB (476K)| 50.3 | **80.6** | 99.7 |88.1|
> |GearNet-Edge-IEConv (Multiview Contras)| 42M |AlphaFoldDB (805K)|54.1 | 80.5 | **99.9** | 87.5 |
> |GearNet-Edge-IEConv (Residue Type Prediction)| 42M|AlphaFoldDB (805K)|48.8 | 71.0 | 99.4 | 86.6|
> |GearNet-Edge-IEConv (Distance Prediction)|42M |AlphaFoldDB (805K)|50.9 | 73.5 | 99.4 | 87.5|
> |GearNet-Edge-IEConv (Angle Prediction)|42M |AlphaFoldDB (805K)|56.5 | 76.3 | 99.6 | 86.8|
> |GearNet-Edge-IEConv (Dihedral Prediction)|42M |AlphaFoldDB (805K)|51.8 | 77.8 | 99.6 |87.0 |
> | ProteinSSA | 100M |-|**60.5** | 79.4 | 99.8 | **89.4** |

---

> ### Author Response · Authors · 2023-11-20
>
> | Method |Param. |Pretraining Dataset|GO-BP | GO-MF | GO-CC | EC |
> | :--- | :---: | :---: | :---: | :---: |:---: |:---: |
> |DeepFRI |6.2M| Pfam (10M)| 0.399 | 0.465 | 0.460 | 0.631 |
> |ESM-1b | 650M|UniRef50 (24M)| 0.470 | 0.657 | 0.488 | 0.864 |
> |ProtBERT-BFD| 420M| BFD (2.1B)| 0.279 | 0.456 | 0.408 | 0.838 |
> |LM-GVP| 420M|UniRef100 (216M)| 0.417 | 0.545 | **0.527** | 0.664|
> | IEConv (amino level)| 36.6M |PDB (476K)| 0.468 | 0.661 | 0.516 |-|
> |GearNet-Edge-IEConv (Multiview Contras)| 42M |AlphaFoldDB (805K)| **0.490** | 0.654 | 0.488 | 0.874 |
> |GearNet-Edge-IEConv (Residue Type Prediction)| 42M |AlphaFoldDB (805K)|0.430 | 0.604 | 0.465 | 0.843|
> |GearNet-Edge-IEConv (Distance Prediction)|42M | AlphaFoldDB (805K)|0.448 | 0.616 | 0.464 | 0.839 |
> |GearNet-Edge-IEConv (Angle Prediction)| 42M |AlphaFoldDB (805K)|0.458 | 0.625 | 0.473 | 0.853 |
> |GearNet-Edge-IEConv (Dihedral Prediction)|42M | AlphaFoldDB (805K)|0.458 | 0.626 | 0.465 | 0.859 |
> |KeAP| 420M |ProteinKG25 (5M)| 0.466 | 0.659 | 0.470 | 0.845 |
> |ESM-2| 650M |UniRef50 (24M)| 0.472 | 0.662 | 0.472 | 0.874 |
> |ProtST-ESM-1b| 759M | ProtDescribe (553K)| 0.480 | 0.661 | 0.488| **0.878** |
> |ProtST-ESM-2| 759M | ProtDescribe (553K)| 0.482 | **0.668** | 0.487| **0.878** |
> | ProteinSSA |100M |-|0.464 | 0.667 | 0.492 | 0.857 |
>
> As shown in these tables, without any pre-training or self-supervised learning, our proposed framework, ProteinSSA, achieves comparable accuracy with those methods with less trainable parameters and even outperforms them on fold and enzyme reaction classification.
>
> **Q3** The GNN architecture is not fully described.
>
> **A3** As mentioned in Section 3.3, a protein sequence is composed of $n$ residues, which are represented as graph nodes. We combine the one-hot encoding of residue types with their physicochemical properties, and they serve as the features for each graph node, denoted as $x_i$. In Section 3.1, we introduced the local coordinate system (LCS) [11] as the geometric features for protein 3D structures. The edge feature $e_{ij}$, defined in Eq. (2), represents the concatenation of the geometric features for protein 3D structures used LCS. We define the sequential distance as $l_{ij} = |i-j|$ and the spatial distance as $d_{ij}=|| P_i-P_j||$, and we establish the edge conditions using Eq. (3). Therefore, the protein's sequential and structural information is captured through the graph node and edge features. In Appendix B.2, we provide detailed information about the graph neural network model. We employ two message-passing layers followed by one average sequence pooling layer. After the pooling layer, the number of residues is halved, and we update the edge conditions before performing another round of message passing and pooling, as illustrated in Figure 1. Thus, the final GNN architecture includes eight message-massing and four pooling layers, which are sufficient for achieving satisfactory results. By leveraging message-passing and pooling layers, we integrate the graphs from different modalities (sequence and structure) into joint embeddings.
>
> **Q4** Can you add the performance of the teacher model into your results tables?
>
> **A4** In terms of the teacher model's performance, as we have stated, we use about 30K proteins with 2752 GO annotations from the GO dataset to train the teacher model, without further division into the biological process (BP), molecular function (MF), and cellular component (CC). The $F_{\mathrm{max}}$ is used as the evaluation metric. We got the $\mathrm{F}_{\mathrm{max}}$ for the teacher model is 0.489 overall.

---

> > ### Author Response · Authors · 2023-11-20
> >
> > Thank you again for all the efforts that helped us improve our manuscript! In case our answers have justifiably addressed your concerns, we respectfully thank you that support the acceptance of our work. As you know, your support holds great significance for us. Also, please let us know if you have any further questions. Look forward to further discussions!
> >
> >
> > [1] Brandes, Nadav, et al. "ProteinBERT: a universal deep-learning model of protein sequence and function." Bioinformatics 38.8 (2022): 2102-2110.
> >
> > [2] Zhou, Hong-Yu, et al. "Protein Representation Learning via Knowledge Enhanced Primary Structure Modeling." bioRxiv (2023): 2023-01.
> >
> > [3] Zeming Lin, Halil Akin, Roshan Rao, Brian Hie, Zhongkai Zhu, Wenting Lu, Allan dos Santos Costa, Maryam Fazel-Zarandi, Tom Sercu, Sal Candido, et al. Language models of protein sequences at the scale of evolution enable accurate structure prediction. BioRxiv, 2022.
> >
> > [4] Gligorijević, Vladimir, et al. "Structure-based protein function prediction using graph convolutional networks." Nature communications 12.1 (2021): 3168.
> >
> > [5] Elnaggar, Ahmed, et al. "Prottrans: Toward understanding the language of life through self-supervised learning." IEEE transactions on pattern analysis and machine intelligence 44.10 (2021): 7112-7127.
> >
> > [6] Zichen Wang, Steven A Combs, Ryan Brand, Miguel Romero Calvo, Panpan Xu, George Price, Nataliya Golovach, Emannuel O Salawu, Colby J Wise, Sri Priya Ponnapalli, et al. Lm-gvp: A generalizable deep learning framework for protein property prediction from sequence and structure.bioRxiv, 2021.
> >
> > [7] Hermosilla, Pedro, and Timo Ropinski. "Contrastive representation learning for 3d protein structures." arXiv preprint arXiv:2205.15675 (2022).
> >
> > [8] Zuobai Zhang, Minghao Xu, Arian Jamasb, Vijil Chenthamarakshan, Aurelie Lozano, Payel Das, and Jian Tang. Protein representation learning by geometric structure pretraining. In International Conference on Learning Representations, 2023.
> >
> > [9] Xu, Minghao, et al. "Protst: Multi-modality learning of protein sequences and biomedical texts." arXiv preprint arXiv:2301.12040 (2023).
> >
> > [10] Rives, Alexander, et al. "Biological structure and function emerge from scaling unsupervised learning to 250 million protein sequences." Proceedings of the National Academy of Sciences 118.15 (2021): e2016239118.
> >
> > [11] John Ingraham, Vikas Garg, Regina Barzilay, and Tommi Jaakkola. Generative models for graph-based protein design. Advances in neural information processing systems, 32, 2019.
> >
> > [12] Anishchenko, Ivan, et al. "De novo protein design by deep network hallucination." Nature 600.7889 (2021): 547-552.
> >
> > [13] Hehe Fan, Zhangyang Wang, Yi Yang, and Mohan Kankanhalli. Continuous-discrete convolution for geometry-sequence modeling in proteins. In The Eleventh International Conference on Learning Representations, 2023.

---

> > > ### Comment · Reviewer_G223 · 2023-11-21
> > >
> > > Thank you for the detailed answers to my questions. I have revised my score accordingly. I have one additional clarification:
> > >
> > > * It seems to be that the performance of ProteinSSA reported in the experiments is that of the student model. 1) Could you clarify this is correct? 2) Could you provide performance numbers for the teacher model within ProteinSSA?

---

> > > > ### Author Response · Authors · 2023-11-21
> > > >
> > > > Dear Reviewer G223,
> > > >
> > > > We genuinely appreciate the time and effort you've dedicated to reviewing our work and your acknowledgment of the answers we've conducted.
> > > >
> > > > **(1)** It seems to be that the performance of ProteinSSA reported in the experiments is that of the student model. Could you clarify this is correct?
> > > >
> > > > **A1** Yes, it is the student model's performance. The training of the teacher model is not our primary concern, which is to learn embeddings in the latent space that contain functional information and provide intermediate supervision for the student model. Our main focus is to obtain comprehensive embeddings for the student model, which is trained using distill loss and task loss without the annotations input.
> > > >
> > > > **(2)** Could you provide performance numbers for the teacher model within ProteinSSA?
> > > >
> > > > (a) Regarding the performance of the teacher model, as mentioned earlier, we trained it using approximately 30K proteins from the GO dataset. We did not separate the annotations into biological process (BP), molecular function (MF), and cellular component (CC). The evaluation metric used was the $F_{\mathrm{max}}$ score. Overall, we achieved a $F_{\mathrm{max}}$ score of 0.489 for the teacher model. (b) If we do not consider the student model, the inputs of the teacher model consist of sequence, structure, and function. However, we currently lack functional data for tasks such as fold classification, enzyme reaction classification, and EC number prediction. Therefore, we evaluated the teacher model using the GO term dataset and calculated $F_{\mathrm{max}}$ scores for BP, MF, and CC. The results are as follows:
> > > >
> > > > | Method| GO-BP | GO-MF | GO-CC |
> > > > | :--- | :---: | :---: | :---: |
> > > > | ProteinSSA (student) | 0.464 | 0.667| 0.492 |
> > > > | ProteinSSA (teacher) | 0.769 | 0.832| 0.849 |
> > > >
> > > > The goal of the Gene Ontology (GO) task is to identify protein functions. Molecular Function (MF) describes activities that occur at the molecular level, with 1,943 classes. Biological Process (BP) represents larger processes, categorized into 489 classes. Cellular Component (CC) describes the parts of a cell or its extracellular environment, with 320 classes [1]. From the provided table, it is evident that incorporating function information as input significantly enhances performance, particularly for MF and CC. These two classes have fewer categories and are more accessible, resulting in higher scores.
> > > >
> > > > [1] Alex Bateman. “UniProt: A worldwide hub of protein knowledge”. In: Nucleic Acids Research (2019).
> > > >
> > > > Thank you again for all the efforts that helped us improve our manuscript! Also, please let us know if you have any further questions. Look forward to further discussions!

---

### Official Review · Reviewer_1uow · 2023-10-30

**Soundness:** 2 fair
**Presentation:** 2 fair
**Contribution:** 2 fair
**Rating:** 5
**Confidence:** 4

**Summary:**

This paper proposes to learn function enhanced protein representations by distilling knowledge from a teacher model with additional GO representation constraint. Here the teacher model is a ProteinBERT, while the GO is encoded by a fully-connected neural network. The combined representation will force the student model to learn meaningful and functional protein representations. The proposed model are evaluated on several understanding tasks, and the performance is pretty good.

**Strengths:**

The proposed model performs well on the several protein understanding tasks.

**Weaknesses:**

1. **Lack of baselines:** The paper lacks some important baselines. For example, the paper didn't report the teacher model's performance and the performance of removing the KL divergence term.

2. **The motivation is unclear:** Actually, I don't really get the reason why the author needed to train a student model, which seems redundant. In this paper, the student model is not smaller than the teacher model. Instead, the student model shares parameters with the teacher model. It seems the author just needs to finetune the ProteinBERT involving the additional GO information constraint.

3. **The writing is confusing:** Many parts of the paper make me feel confused, especially the KL divergence part. For example, what do $P_S(G_S, A)$ and $P(Z_S|G_S, A)$ mean? Are these VAE model? If it's true, then expanding the $P_S(G)$ to $P(G|z)P(z)$, I don't think the assumption that "$E_{p_S(G)}[KL[p_S(y|G)P_T()y|G]]$ does not depend on z" holds. By the way, I don't really get what the source domain and target domain mean. It seems they are the same domain in the exception that source has an additional constraint on GO.

**Questions:**

I have already mentioned some questions in the weaknesses. Additional questions are provided as follows:

1. In Equation 5, why the author directly add the $h_S$ to h_A without any transformation? It seems they are from different semantic spaces.

2. Removing the AE-T doesn't influence the performance much. Does that mean this additional GO encoder didn't add to much benefit to the whole model?

---

> ### Author Response · Authors · 2023-11-20
>
> Dear Reviewer 1uow,
>
> We are grateful for your thorough review. Your comments are highly valued, and we would like to express our heartfelt gratitude. We do our utmost to address the questions you have raised:
>
> **Q1** The paper lacks some important baselines. For example, the paper didn't report the teacher model's performance and the performance of removing the KL divergence term.
>
> **A1** Thank you for your valuable feedback! (1) The proposed method is compared with existing popular protein representation learning methods. We have reported the performance of removing the KL divergence term in the ablation study, shown in Table 5, w/o Teacher means removing the teacher model, which leads to substantial performance drops across all tasks compared to the full ProteinSSA. (2) In terms of the teacher model's performance, as we have stated, we use approximately 30K proteins from the GO dataset to train the teacher model, without further division into the biological process (BP), molecular function (MF), and cellular component (CC). The $F_{\mathrm{max}}$ is used as the evaluation metric.  Overall, we achieved a $F_{\mathrm{max}}$ score of 0.489 for the teacher model.  (3) If we do not consider the student model, the inputs of the teacher model consist of sequence, structure, and function. Therefore, we evaluated the teacher model using the GO term dataset and calculated $F_{\mathrm{max}}$ scores for BP, MF, and CC. The results are as follows:
>
> | Method| GO-BP | GO-MF | GO-CC |
> | :--- | :---: | :---: | :---: |
> | ProteinSSA (student) | 0.464 | 0.667| 0.492 |
> | ProteinSSA (teacher) | 0.769 | 0.832| 0.849 |
>
> The goal of the Gene Ontology (GO) task is to identify protein functions. Molecular Function (MF) describes activities that occur at the molecular level, with 1,943 classes. Biological Process (BP) represents larger processes, categorized into 489 classes. Cellular Component (CC) describes the parts of a cell or its extracellular environment, with 320 classes. From the provided table, it is evident that incorporating function information as input significantly enhances performance, particularly for MF and CC. These two classes have fewer categories and are more accessible, resulting in higher scores.
>
> **Q2** Actually, I don't really get the reason why the author needed to train a student model, which seems redundant. In this paper, the student model is not smaller than the teacher model. Instead, the student model shares parameters with the teacher model. It seems the author just needs to finetune the ProteinBERT involving the additional GO information constraint.
>
> **A2** Thank you for your informative reviews! (1) At present, not even 1% of sequenced proteins have functional annotation [1,2]. While the teacher network requires extra functions as input, such information is not always available. As discussed in Section 3.2, we highlight the limitations of pre-training and the absence of a well-learned protein functional encoder to encode functional information. To address these challenges and make better use of functional information without extensive pre-training, we propose the multimodal knowledge distillation framework. (2) Compared with the teacher model, the student model is smaller than the teacher model, due to the deprecation of the annotation encoder. (3) The student model is trained by a hybrid loss, present in Eq.(10) in the revised manuscript, it does not share parameters with the teacher model. The Kullback-Leibler (KL) divergence loss is to regularize the parameters of the student model indirectly through the distribution of its embeddings. (4) ProteinBERT [3] is the first model to utilize the protein annotation information, using 8943 frequent GO annotations. The current state-of-the-art (SOTA) sequence-function pre-trained model is the KeAP [4], but the SOTA model KeAP fails to generate protein embeddings that surpass those obtained from the sequence pre-trained model ESM-1b [5]. This observation demonstrates the limitations of the sequence-function pre-trained model and motivates the development of ProteinSSA. (5) In Section 3.2, we have included a preliminary experiment to highlight the limitations of existing sequence-function pre-training models. This experiment serves as the motivation for proposing the multimodal knowledge distillation framework, ProteinSSA, which aims to embed protein sequence, structure, and function.
>
> **Q3** Many parts of the paper make me feel confused, especially the KL divergence part. For example, what do $P_S(G_S,A)$ and $P(Z_S|G_S,A)$ mean? Are these VAE model? If it's true, then expanding the $P_S(G)$ to $P(G|z)P(z)$, I don't think the assumption that "$E_{p_S(G)}\left[K L\left[p_S(y \mid G) P_T() y \mid G\right]\right]$ does not depend on z" holds. By the way, I don't really get what the source domain and target domain mean. It seems they are the same domain in the exception that source has an additional constraint on GO.

---

> ### Author Response · Authors · 2023-11-20
>
> **A3** (1) In the context of domain adaptation, a domain refers to a specific distribution of data. The main goal is to overcome the distribution shift between the source and target domains, which can lead to a decrease in performance when directly applying a model trained on the source domain to the target domain. (2) In section 3.3, we have clarified that the reason for employing domain adaptation strategies is to develop a sample-independent method, particularly for protein samples without available functional labels. The goal is to align the student's latent distributions obtained from sequence-structure embeddings to the distributions of the teacher model's latent space, thereby bridging the gap between different domains and enhancing the performance of the student model. The source domain is the distribution of teacher model's embeddings, and the target domain is the distribution of student model's embeddings. (3) $P(Z_S|G_S,A)$ mean the distribution of $z_S$ in the source domain. In Section 3.1, we have given the definitions. There is a source domain $S$ with the data distribution $p_S(z_S| G_S, A)$, and there is also a target domain $T$ with the data distribution $p_T(z_T|G_T)$, the latent embeddings $z_S, z_T$ are from the teacher and student network for a protein. $A$ with $k$ terms are the GO annotations. $G_S, G_T$ are the input graphs in the source and target domains. Thus, $P_S(G_S,A)$ means the distribution of the input graph $G_S$ and annotation $A$. (4) This is not a VAE model. (5) In Eq.(8) in the revised manuscript, it defines the misalignment in the representation space between the source and target domains. The theoretical derivations are shown in Appendix D. $E_{p_S(G)}\left[KL\left[p_S(y|G)\parallel p_T(y|G)\right]\right]$ is fixed and and often small [7] (not dependent on the representation $z$, and $y$ is the protein class), there is no $z$ in this term, which is not dependent on the representation $z$. From Proposition 2 shown in Appendix D, we can see the conditional misalignment in the representation space is bounded by the conditional misalignment in the input space.
>
> **Q4** In Equation 5, why the author directly add the $h_S$ to $h_A$ without any transformation?
>
> **A4** The annotations associated with $G_S$ serve as the input for the annotation encoder, resulting in the extraction of feature vector $h_A$. Notably, both $h_S$ (the feature vector obtained from processing $G_S$) and $h_A$ possess identical feature dimensions, enabling a direct addition of these vectors.
>
> **Q5** Removing the AE-T doesn't influence the performance much.
>
> **A5** (1) In our study, we conducted ablation experiments to analyze the impact of excluding the annotation encoder in the teacher model. This is denoted as "w/o AE-T" in Table 5. The results showed a slight decrease in performance because function information was incorporated into the loss function without using the annotation encoder. However, completely removing the teacher model (w/o Teacher) resulted in significant performance drops across all tasks compared to the full ProteinSSA approach. These ablations highlight the importance of utilizing the annotation encoder and incorporating function information in the loss function and the teacher model for optimal results. (2) By removing the annotation encoder in the teacher model but still incorporating function information into the loss function, it can be viewed as a multi-task learning method. This approach leverages knowledge transfer across tasks, leading to improved generalization performance. (3) The auxiliary annotation encoder in the teacher model is implemented as a multi-layer perceptron. Further details about the model can be found in Appendix B.2. Additionally, the GO encoder can be enhanced by utilizing large language models or graph neural networks, as well as incorporating more GO terms to capture a broader range of functional information.
>
> Thank you again!
>
> [1] Torres, M., Yang, H., Romero, A. E. & Paccanaro, A. Protein function prediction for newly sequenced organisms. Nat. Mach. Intell. 3, 1050–1060 (2021).
>
> [2] Ibtehaz, et al. "Domain-PFP allows protein function prediction using function-aware domain embedding representations." Communications Biology 6.1 (2023): 1103.
>
> [3] Brandes, Nadav, et al. "ProteinBERT: a universal deep-learning model of protein sequence and function." Bioinformatics 38.8 (2022): 2102-2110.
>
> [4] Hong-Yu Zhou, et al. Protein representation learning via knowledge enhanced primary structure modeling, 2023.
>
> [5] Zeming Lin, et al. Language models of protein sequences at the scale of evolution enable accurate structure prediction. BioRxiv, 2022.
>
> [6] Hehe Fan, et al. Continuous-discrete convolution for geometry-sequence modeling in proteins. In The Eleventh International Conference on Learning Representations, 2023.
>
> [7] Ben-David, Shai, et al. "A theory of learning from different domains." Machine learning 79 (2010): 151-175.

---

> > ### Comment · Reviewer_1uow · 2023-11-22
> > **Reply to the rebuttal**
> >
> > Thanks for the response. I have carefully read the author's response and other reviewers' comments. I do agree with reviewer KkPB's opinion that there are too many elements incorporated into this method without a cohesive story, making the paper confusing. For example, I still feel unclear about the motivation of training a student model instead of directly finetuning ProteinBERT with additional functional constraints. Therefore, I'd like to keep my current score.

---

> ### Author Response · Authors · 2023-11-23
>
> Dear Reviewer 1uow,
>
> We genuinely appreciate the time and effort you've dedicated to reviewing our work and your acknowledgment of the answers we've conducted.
>
> **(1)** I have carefully read the author's response and other reviewers' comments. I do agree with reviewer KkPB's opinion that there are too many elements incorporated into this method without a cohesive story, making the paper confusing. For example, I still feel unclear about the motivation of training a student model instead of directly finetuning ProteinBERT with additional functional constraints. Therefore, I'd like to keep my current score.
>
> **A1** Thanks for your criticisms. We have made significant efforts to improve the manuscript and have just completed the latest round of revisions. The submitted version has undergone multiple iterations, with each revision incorporating valuable feedback and making necessary improvements.
>
>  At present, not even 1% of sequenced proteins have functional annotation [1,2]. While the teacher network requires extra functions as input, such information is not always available. As discussed in Section 3.2, we highlight the limitations of pre-training and the absence of a well-learned protein functional encoder to encode functional information. To address these challenges and make better use of functional information without extensive pre-training, we propose the multimodal knowledge distillation framework.
>
> In case our answers have justifiably addressed your concerns, we respectfully thank you that support the acceptance of our work. As you know, your support holds great significance for us. Also, please let us know if you have any further questions. Look forward to further discussions!
>
> [1] Torres, M., Yang, H., Romero, A. E. & Paccanaro, A. Protein function prediction for newly sequenced organisms. Nat. Mach. Intell. 3, 1050–1060 (2021).
>
> [2] Ibtehaz, Nabil, Yuki Kagaya, and Daisuke Kihara. "Domain-PFP allows protein function prediction using function-aware domain embedding representations." Communications Biology 6.1 (2023): 1103.

---

> > ### Comment · Reviewer_1uow · 2023-11-23
> > **Further response**
> >
> > I appreciate the authors' effort in revising the manuscript. I will read the updated version shortly and then see if I have the motivation to update my current score.

---

> > > ### Author Response · Authors · 2023-11-23
> > >
> > > Thank you again for all the efforts that helped us improve our manuscript.  Look forward to further discussions.

---

> > > > ### Comment · Reviewer_1uow · 2023-11-23
> > > > **Follow-up comment**
> > > >
> > > > I would like to read the updated version of the manuscript. However, I didn't find where the revised parts are. Maybe the author needs to highlight the revised part to let us know the main revision of the paper. Considering the the effort the author have put into the rebuttal process, I raise my score from 3 to 5, but I still think there is a lot of space for this paper to be improved.

---

> > > > > ### Author Response · Authors · 2023-11-23
> > > > >
> > > > > Dear Reviewer 1uow,
> > > > >
> > > > > Thanks very much for your patience and efforts in our work! We are grateful for your review. Your comments are highly valued, and we would like to express our heartfelt gratitude, we learned a lot from them.
> > > > >
> > > > > Considering the locations of the revised parts, we have highlighted them with red color in the updated version.
> > > > >
> > > > > Thanks again!  We respectfully thank you that support the acceptance of our work. Also, please let us know if you have any further questions. Look forward to further discussions!
> > > > >
> > > > > Best regards,
> > > > >
> > > > > Authors.

---

### Official Review · Reviewer_KkPB · 2023-10-31

**Soundness:** 3 good
**Presentation:** 2 fair
**Contribution:** 3 good
**Rating:** 6
**Confidence:** 5

**Summary:**

This paper is concerned with how to compute embedded representations of proteins using a variety of data sources.  The authors note the imbalanced nature of protein data, where unannotated sequences are plentiful, with functional annotations an order of magnitude less so, and structures rarer by yet another order of magnitude.  Their solution is to fuse representations by distilling knowledge from a teacher network, for which structure, sequence, and annotations exist, and a student network that acts on sequence and structure alone.  While the student and teacher share a GNN architecture for encoding sequence and structure, the teacher additionally has function annotation information to enrich its' embeddings.  The teacher's richer embedding space regularizes the student's embedding space, thus imparting the student with extra information.  They go on to show favourable performance on tasks predicting fold classification, enzyme reaction, and GO-term predictions.

**Strengths:**

- The authors' attempt to circumvent the data imbalance for annotated protein data by using distillation is really interesting, and I think is well worth exploring here.
- In addition, combining the best of 1D, 3D (via a replacement for SE(3) tools via [Ingraham et al.]() and GNNs is really interesting too.
- In particular, the result for ProteinSSA on fold classification is clearly an advance.

**Weaknesses:**

The largest weakness of this paper isn't in the ideas, but with the writing.  There are many examples where it's not quite clear from the text what the authors are trying to convey.

- The authors stress that ProteinSSA does not require pretraining, and does not make use of annotations.  This is only true of the *student* model, since the teacher does clearly make use of annotations where such exist. Table 1 is thus misleading.

- The title of section 3.2 isn’t very informative.  What problem does it address?  Or what part of the final architecture is being discussed here?  It’s not clear, and would benefit from being rewritten.  For example, how does CDConv relate to KeAP and ESM-1b?  These are very briefly described, but not in sufficient detail to tell the reader why each was chosen, and how they relate to each other & to ProteinSSA as a whole. It’s only when you read to the bottom of 3.2 that you discover that this section is all about establishing that pre-trained models are limited in different ways, and *that’s* why ProteinSSA was made.  Please, lead with this, and then describe the limitations of other sequence-based models that require extensive pre-training afterwards.

- Reading through the subsections of section 3, it’s hard to put my finger on what the focus of this paper is.
The different elements are well described, but what isn’t clear exactly is how they will be synthesized into something new and exciting, as well as why the choices (e.g edge representation of 3.1, sequence function representation of 3.2) were made, and why (beyond them being SOTA at one point in time).  I think section 3 would be clearer, and benefit the reader if it had a summary of the subsections at the beginning, and for each subsection to describe one part of the whole model

- Reading through to Section 3.3, it isn't yet explained why the authors think knowledge distillation is the best way to incorporate knowledge from annotations. Why not just use the teacher network directly? The answer is (I think from subsequent sections) that functional annotation are only sometimes available, so by instead aligning the latent space of the student with that of the teacher, the student derives the benefit of additional knowledge.  I have to stress this is not clear from reading section 3, but should be clearly spelled out somewhere within (or within the introduction).

- Both the sentence preceding equation 8 and the sentence that follow are overly wordy, but without the benefit of clarity.  It’s clear that the addition of a KL regularization term KL($P_{S}(z_{S})$, $P_{T}(z_{T})$) will force the student embedding distribution to become like the teacher distribution, thus affecting the student embedding state.  Words about “reduce the bound in the representation spaces” or “KL divergence matches distributions…” is a bit misleading; all that’s intended here is an intention to regularize the parameters of the student  model indirectly through the distribution of its embeddings.

- Table 3 reports only point estimates of max accuracy.  I find max accuracies very difficult to parse in a meaningful way.  I think the improvements of ProteinSSA would be better qualified if you report the distribution of accuracies from multiple runs, especially that of fold classification.  Even if you cannot re-run the alternatives, you can report ProteinSSA results more faithfully.


**Minor points:**

In the introduction, the phrase ‘grammar of life’ isn’t a helpful metaphor.  I realize this is a small point, but what these models learn are not always distillable into rules for compositional orientation of elements of protein language.

- Equation 5 has a term $\alpha$ that controls "the isotropic of protein representations".  What does this mean?
- There are some grammtical errors in the first sentence on page 6
- Page 6 in section 3.3 invokes the CLT.  I don’t think you need to invoke the CLT here to model the distribution of the embeddings as Gaussian, you can just assume it to be true.  At any rate, it’s not clear that the different batch derived embeddings are independent.

**Questions:**

- The ablation study of section 4.5 is welcome, but does not address one of the key choices of the paper (raised in the *Protein Domain Adaptation* paragraph of section 3.3), which is why the teacher embeddings are concatenated with a separate functional embedding rather than using function as an extra term in the loss function for classification.  How come?

- Just prior to equation 10, there is an argument about reducing the generalization bound which seems a non-sequitur.  I do not understand why generalization bound arguments are being used here; it seems very disconnected from the rest of this section.  Could the authors please help me understand why?

- Section 4.1 begins by describing ProteinBERT and how it is pre-trained.  Is this part of ProteinSSA?  If so, can ProteinSSA really claim (as in table 1) that it is not pre-trained?  If not, then is mentioning ProteinBERT here relevant?


I want to stress to the authors that I think there is a good paper within here, but that its writing needs work, and that the authors need to think harder about ordering, motivating, and presenting their arguments.  I'm certainly willing to change my score if the post-rebuttal version of the paper takes my suggestions into account.

---

> ### Author Response · Authors · 2023-11-20
>
> Dear Reviewer KkPB,
>
> Thanks for your appreciation and detailed review. We respond to your comments with a heart full of gratitude. We try our best to response the questions below:
>
> **Q1** ProteinSSA does not require pretraining, and does not make use of annotations. This is only true of the student model. Table 1 is thus misleading.
>
> **A1** Thank you for your valuable feedback! (1) It is indeed true that ProteinSSA uses annotations for the teacher model, its objective is to learn embeddings in the latent space that contain functional information and provide intermediate supervision during knowledge distillation for the student model. Therefore, the complete training of the teacher model is not our primary concern. Our main focus is to obtain comprehensive embeddings for the student model, which is trained using distillation loss and task loss without the annotations input. (2) Pre-training involves learning general representations from a large dataset, while fine-tuning adapts the pre-trained model to a specific task or dataset. In the teacher-student framework, the teacher model is usually a well-learned model that serves as a source of knowledge for the student model. The student model aims to mimic the behavior or predictions of the teacher model. In this work, the teacher model is trained on a relatively small dataset consisting of about 30 thousand proteins with 2752 GO annotations. This is in comparison to ProteinBERT [1], which uses 106 million proteins, and KeAP [2], which uses 5 million triplets. It can still be considered as training rather than pre-training for the teacher model, and we mainly want to obtain a better student model. We have clarified this information and modified Table 1 in the revised version for better understanding.
>
> **Q2** What problem does section 3.2 address? Or what part of the final architecture is being discussed here? It’s not clear, and would benefit from being rewritten. How does CDConv relate to KeAP and ESM-1b? These are very briefly described, but not in sufficient detail to tell the reader why each was chosen, and how they relate to each other & to ProteinSSA as a whole. It’s only when you read to the bottom of 3.2 ... , lead with this, and then describe the limitations of other sequence-based models that require extensive pre-training afterwards.
>
> **A2** Thank you for your informative reviews! We have carefully considered your suggestions and made the following revisions to the manuscript. (1) In Section 3.2 of the revised manuscript, we now begin with a summary and central idea of the subsection, followed by the preliminary experiment. We emphasize that the performance of pre-trained models is influenced by various factors, such as model size, dataset scale, and choice of pre-training tasks. To illustrate this, we have conducted an experiment. (2) In this experiment, we want to highlight that the current state-of-the-art (SOTA) sequence-function pre-trained model, KeAP [2], fails to generate protein embeddings that surpass those obtained from the sequence pre-trained model ESM-1b [3]. This observation demonstrates the limitations of the sequence-function pre-trained model and motivates the development of ProteinSSA. (3) As mentioned in Section 3.2, CDConv [4] introduces an effective fundamental operation to encapsulate protein structure and sequence without pre-training or self-supervised learning. CDConv achieves superior performance compared to pre-training methods like GearNet-Edge-IEConv [5]. CDConv is currently the most effective publicly available method for modeling protein sequence and structure. In the field of protein pre-training, KeAP is the current SOTA model for downstream tasks, pre-trained on protein sequences and functions, while ESM-1b is the most prevalent sequence pre-training model. By incorporating the embeddings from KeAP and ESM-1b to enhance the embeddings obtained from CDConv, we can compare the quality and performance of the embeddings from the two pre-trained models, evaluating their ability using the solid sequence-structure base model, CDConv. (4) The student model in the ProteinSSA framework is designed to model protein sequence and structure together, learning functional information from the teacher model without pre-training. Additionally, the message-passing mechanism formulated in Eq.(4) is inspired by CDConv, which convolves node and edge features from sequence and structure simultaneously.

---

> > ### Comment · Reviewer_KkPB · 2023-11-21
> > **response to authors' comment**
> >
> > Thanks for your reply, I'm encouraged to read that the motivation of ProteinSSA has been addressed (and contextualized) with more care.
> >
> > > Pre-training involves learning general representations from a large dataset, while fine-tuning adapts the pre-trained model to a specific task or dataset. In the teacher-student framework, the teacher model is usually a well-learned model that serves as a source of knowledge for the student model. The student model aims to mimic the behavior or predictions of the teacher model. In this work, the teacher model is trained on a relatively small dataset consisting of about 30 thousand proteins with 2752 GO annotations. This is in comparison to ProteinBERT [1], which uses 106 million proteins, and KeAP [2], which uses 5 million triplets. It can still be considered as training rather than pre-training for the teacher model, and we mainly want to obtain a better student model.
> >
> > I'm not convinced that the distinction between training and pre-training is about the size of the dataset.  If you're beginning from a random initialization and are updating model weights according to some supervised loss, I think most people would consider that training.  Pre-training is usually, as the authors point out, performed using a surrogate (often self-supervised or unsupervised) loss on a much larger data set, and then a pre-trained model is fine-tuned using a more relevant objective and target dataset.  I guess what I'm most unhappy about is that the details of the teacher model aren't really the focus here; the teacher model is constructed so that the distilled student model can be as powerful as possible, and so any teacher model that admits fine-tuning (or training) on function annotations should suffice.
> >
> > While I think that section 3.2 and 3.3 do a better job of motivating ProteinSSA, I think there's still more attention that could be paid to the writing.  For instance, when motivating why the distributions of the latent spaces between the teacher model and student model are aligned
> >
> > > Importantly, rather than directly minimizing distances between sample-dependent embeddings ZS and ZT , we develop a sample-independent method. This aligns the student’s latent space with the teacher’s by approximating the distributions of sequence-structure embeddings to the triplet representations. This distribution alignment approach avoids reliance on individual sample embeddings.
> >
> > what isn't mentioned here are the motivations for **why** it's helpful to decouple the teacher model and the student model.  One important reason is that the teacher model can then be trained on a much larger dataset, or multiple datasets, without requiring such information to be available to the student model.  This may seem small, but emphasizing these details helps to establish the motivation.

---

> > > ### Author Response · Authors · 2023-11-22
> > >
> > > Dear Reviewer KkPB,
> > >
> > > We sincerely value the time and energy you have devoted to assessing our work and recognizing the efforts we have put into providing responses.
> > >
> > >
> > > **(1)** I'm not convinced that the distinction between training and pre-training is about the size of the dataset. I guess what I'm most unhappy about is that the details of the teacher model aren't really the focus here; the teacher model is constructed so that the distilled student model can be as powerful as possible, and so any teacher model that admits fine-tuning (or training) on function annotations should suffice.
> > >
> > > **A1** Thanks for your valuable comments! (a) I agree with the point that pre-training refers to the phase of training a machine learning model using a large dataset in an unsupervised or self-supervised manner. As mentioned in the last paragraph of Section 3.3, the teacher model is indeed trained using a supervised cross-entropy loss, denoted as $L_{\mathrm{teacher}}$. This loss function allows the teacher model to learn from labeled data and optimize its parameters accordingly. (b) The teacher model consists of well-designed graph neural networks that are capable of obtaining sequence-structure embeddings. Additionally, it includes a functional encoder that captures functional embeddings. In the teacher-student framework, the teacher model serves as a valuable source of knowledge and guidance for the student model. Typically, the teacher model is more complex or larger in size, as it has been extensively trained on a protein sequence-structure dataset with functional annotations. This enables the teacher model to capture intricate patterns and relationships between sequences, structures, and functions.
> > >
> > > **(2)** When motivating why the distributions of the latent spaces between the teacher model and student model are aligned.
> > >
> > > **A2** Thanks for your insightful reviews! There are several reasons for aligning the distributions of the latent spaces between the teacher and student models. Firstly, in our approach, we obtain protein embeddings in the latent spaces of both the teacher and student models. As mentioned in Equation 6, the teacher model is trained on functional datasets that contain embeddings reflecting patterns from the protein sequence, structure, and function. To transfer this knowledge effectively, we align the distribution of embeddings between the teacher and student models. Secondly, instead of directly aligning the individual embeddings, we choose to align the distributions of these embeddings. As discussed in Section 3.3, the protein embeddings can be sample-dependent, meaning they require the same protein input for both the teacher and student networks. However, our goal is to develop a sample-independent method. By aligning the distribution of embeddings, we avoid relying on individual sample embeddings. The distribution alignment approach, represented by $p_S(z_S)$, can be seen as obtained from the entire set of function-annotated protein samples. We appreciate your valuable input, and we have emphasized these points in the revised manuscript to provide a clearer understanding of why we align the distributions of the latent spaces rather than directly aligning the individual embeddings.
> > >
> > >
> > > **(3)** Why it's helpful to decouple the teacher model and the student model.
> > >
> > > **A3** Thanks for your careful reviews! There are several compelling reasons for decoupling the teacher model from the student model. Firstly, as previously mentioned, the teacher model can be trained on a larger dataset or multiple datasets, enabling it to benefit from a broader range of information without requiring the same level of access for the student model. This allows the teacher model to capture more complex patterns and generalize better. Secondly, it is important to consider the limited availability of functional annotations for proteins. As stated, only a small fraction of sequenced proteins currently have functional annotations [1]. Therefore, relying solely on function annotations as input for the teacher network may severely limit its applicability. By decoupling the models, the student model can still learn from the teacher model's knowledge without explicitly requiring such annotations. Lastly, the student model is typically designed to be smaller and more lightweight than the teacher model. This makes it easier to deploy and utilize in various scenarios. By mimicking the behavior and performance of the teacher model, the student model can effectively leverage the knowledge distilled from the teacher while maintaining efficiency. By considering these factors, we can see that decoupling the teacher and student models offers advantages in terms of leveraging larger datasets, accommodating limited functional annotations, and ensuring practicality and convenience in deployment. These considerations further motivate the use of a separate student model in the teacher-student framework.

---

> ### Author Response · Authors · 2023-11-20
>
> **Q3** The different elements are well described, but what isn’t clear exactly is how they will be synthesized into something new and exciting, as well as why the choices (e.g edge representation of 3.1, sequence function representation of 3.2) were made, and why (beyond them being SOTA at one point in time).
>
> **A3** Thank you for your suggestions! We have carefully reviewed and incorporated your feedback into the manuscript, especially for section 3, we add a summary of the subsections at the beginning, and describe what they functions. For example, (1) we define the problem and notations, and introduce the local coordinate system (LCS) [7] as the geometric features for protein 3D structures. LCS features are widely adopted in protein structure modeling methods [4,8], as they provide informative and comprehensive descriptions of protein structures by considering distance, direction, and orientation. (2) In Section 3.2, we have included a preliminary experiment to highlight the limitations of existing sequence-function pre-training models. This experiment serves as the motivation for proposing the multimodal knowledge distillation framework, ProteinSSA, which aims to embed protein sequence, structure, and function. (3) In Section 3.3, we have presented the overall framework of ProteinSSA, which involves training a teacher model and a student model through iterative knowledge distillation. We have provided a detailed explanation of the network architecture, highlighting the use of the same Graph Neural Network (GNN) architecture in both the teacher and student models. Additionally, we have introduced the concept of protein domain adaptation and derived the loss for the student model. The message-passing mechanism used in both the teacher and student models is inspired by CDConv, which enables better modeling of protein sequences and structures. We have clarified that the reason for developing a sample-independent method, for proteins without available functional labels. The goal is to align the student's latent distributions obtained from sequence-structure embeddings to the distributions of the teacher model's latent space, thereby bridging the gap between different domains and enhancing the performance of the student model.
>
> **Q4** It isn't yet explained why the authors think knowledge distillation is the best way to incorporate knowledge from annotations. Why not just use the teacher network directly?
>
> **A4** Thank you for your reviews! At present, not even 1% of sequenced proteins have functional annotation [9,10]. While the teacher network requires extra functions as input, such information is not always available. Another way is the pre-training and fine-tuning paradigm, as discussed in Section 3.2, we highlight the limitations of pre-training and the absence of a well-learned protein functional encoder to encode functional information. To address these challenges and make better use of functional information without extensive pre-training, we propose the multimodal knowledge distillation framework.
>
> **Q5** Both the sentence preceding equation 8 and the sentences that follow are overly wordy, but without the benefit of clarity. Words about “reduce the bound in the representation spaces” or “KL divergence matches distributions…” is a bit misleading;
>
> **A5** Thank you for your informative reviews! We appreciate your feedback and have taken steps to clarify wordy sentences. Regarding Eq.(6)-Eq.(10), we would like to clarify that these equations were used to illustrate the process of deriving an ideal loss for the student model in a theoretical manner, where the goal is to align the student embedding distribution with the teacher embedding distribution. We apologize for any confusion caused and have made the necessary rectifications.
>
> **Q6** Table 3 reports only point estimates of max accuracy.
>
> **A6** Thank you for highlighting this aspect! In Section 4.1, we present the performance is measured with averaged values over three initializations, this means that we report the mean terms of three independent runs instead of point estimation on these four tasks, including protein fold classification, enzyme reaction classification, Gene Ontology (GO) term prediction, and enzyme commission (EC) number prediction. For the variance or confidence intervals of reported results, we follow the results of baselines reported in [4]. For the task of fold and reaction classification, the performance is measured as mean accuracy. For GO Term and EC number prediction, the $\mathrm{F}_{\mathrm{max}}$ is used as the evaluation metric. Here, we report the mean (variance) for the proposed ProteinSSA:
>
> | Method | Fold | SuperFamily | Family | Enzyme Reaction | GO-BP | GO-MF | GO-CC | EC |
> | :--- | :---: | :---: | :---: | :---: | :---: | :---: | :---: | :---: |
> | ProteinSSA | 60.5(0.60) | 79.4(0.96) | 99.8(0.04) | 89.4(0.43) | 0.464(0.007) | 0.667(0.003) | 0.492(0.004) | 0.857(0.011) |

---

> ### Author Response · Authors · 2023-11-20
>
> **Q7** The phrase ‘grammar of life’ isn’t a helpful metaphor.
>
> **A7** Thank you for your comment! We highly appreciate your feedback and acknowledge that the knowledge acquired by these models cannot always be distilled into explicit rules for the compositional orientation of elements in the protein language. In the introduction, we discuss protein language models (PLMs) that have demonstrated the ability to learn the 'grammar of life' from large numbers of protein sequences. In works such as [12,13], language models are applied to model protein sequences, highlighting the common characteristics shared between human languages and proteins. For instance, the hierarchical organization [14] suggests that the four levels of protein structures can be analogized to letters, words, sentences, and texts in human languages to a certain degree. In light of these considerations, it may be appropriate to modify the phrase "learn the grammar of life" to "learn the certain grammar of life".
>
> **Q8** Equation 5 has a term $\alpha$ that controls "the isotropic of protein representations". What does this mean?
>
> **A8** Thank you for your question! In this context, the term "isotropic" refers to the uniformity or symmetry of protein representations. It may be better to change the word isotropic to balance.  The hyper-parameter $\alpha$ controls the balance between the contribution of the annotation encoder ($h_A$) and the graph-level protein embeddings ($h_S$) in the combined representation ($z_S$). By adjusting the value of $\alpha$, we can control the extent to which the two components contribute to the final representation. A higher value of $\alpha$ would give more weight to the graph-level protein embeddings, while a lower value would amplify the influence of the annotation encoder. This allows us to control the isotropy or symmetry of the resulting protein representations, determining whether they are more influenced by the annotation or the structural information captured in the graph-level embeddings.
>
> **Q9** A few minor typos in the first sentence on page 6.
>
> **A9** Thanks for your careful reviews! We sincerely appreciate your detailed and valuable suggestions, and we thoroughly revise our paper based on your constructive comments.
>
> **Q10** I don’t think you need to invoke the CLT here to model the distribution of the embeddings as Gaussian, you can just assume it to be true.
>
> **A10** Thank you for your suggestion! It is true that we can assume protein embeddings follow a Gaussian distribution without mentioning the central limit theorem [15]. We have modified this in the revised version.
>
> **Q11** why the teacher embeddings are concatenated with a separate functional embedding rather than using function as an extra term in the loss function for classification. How come?
>
> **A11** Thank you for insightful reviews! (1) We conducted ablation experiments, shown in Table 5, where we excluded the annotation encoder in the teacher model, denoted as "w/o AE-T" in Table 5. This means that we incorporated function information into the loss function for the teacher models, while deprecating the use of the annotation encoder. As a result, there was a slight decrease in performance observed. These ablations demonstrate the importance of using the annotation encoder and incorporating function information in the loss function for optimal results. (2) The input functions used in the annotation encoder can be considered as part of the Label-Augmented method. Previous works have successfully utilized label-augmented techniques to enhance model training [16,17]. This technique involves encoding labels and combining them with node attributes through concatenation or summation. By doing so, it improves feature representation and enables the model to effectively utilize valuable information from labels. The Label-Augmented method is applicable to various types of graphs and can be employed in different domains. Thus, we choose to incorporate functional embeddings, concatenated with graph-level protein sequence-structure embeddings. (3) Another approach to improve the model's performance is by incorporating function as an additional term in the loss function, which is commonly seen in multi-task learning (MTL) methods. The objective of MTL is to leverage knowledge transfer across tasks and achieve better generalization. However, when tasks are diverse, a simple shared MTL model can suffer from task interference [18]. Previous work [19] has explored the relationship and mutual adaptability of protein tasks. Considering the concept of prompt learning [18] in the MTL domain, it holds promise as a potential direction for improvement.
>
> | Method | Fold | SuperFamily | Family | Enzyme Reaction | GO-BP | GO-MF | GO-CC | EC |
> | :--- | :---: | :---: | :---: | :---: | :---: | :---: | :---: | :---: |
> | ProteinSSA | 60.5 | 79.4 | 99.8 | 89.4 | 0.464 | 0.667 | 0.492 | 0.857 |
> | w/o AE-T | 60.4 | 79.1 | 99.7 | 88.9 | 0.454 | 0.664 | 0.490 | 0.854 |

---

> > ### Comment · Reviewer_KkPB · 2023-11-21
> > **Response to authors**
> >
> > Regarding the authors answer to Q8 ('isotropic'), this description that the authors provide I think is the clearest description of what the $\alpha$ parameter controls, and maybe should be what appears in the text:
> >
> > > The hyper-parameter $\alpha$ controls the balance between the contribution of the annotation encoder ($h_{A}$) and the graph-level protein embeddings ($h_{S}$) in the combined representation ($z_{S}$).
> >
> > Regarding the choice to incorporate functional annotations through label encoding rather than as a separate loss function, I appreciate the justification via Bengio et al and Sun et al.  I think this justification should also appear in the text (and please add both citations to the manuscript).

---

> ### Author Response · Authors · 2023-11-20
>
> **Q12** Why generalization bound arguments are being used here;
>
> **A12** Thanks for your valuable comments. There exists a misalignment for KL-guided domain adaptation method in the representation space between source domain and target domain because of data differences. Theoretically, the student model should have a lower generalization bound when transferring knowledge from the teacher model, as shown in Eq.(8) in the revised manuscript.  In order to approximate the ideal target domain loss, we reduce the terms inside the radical sign, because the second term in Eq.(8) is often small and fixed.  Thus, we only need to leverage Kullback-Leibler (KL) divergence to align the distributions between source domain and target domain, i.e., $KL[p_S(z)|| p_T(z)]$.
>
> **Q13** Section 4.1 begins by describing ProteinBERT and how it is pre-trained. Is this part of ProteinSSA?
>
> **Q13** Thanks for your question! ProteinBERT [1] is not a part of ProteinSSA. It is the first model to utilize the protein annotation information, using 8943 frequent GO annotations. Compared with ProteinBERT, we use 2752 GO annotations from the GO dataset. We have modified this in the revised manuscript to decrease the misleading.
>
> Thank you again for all the efforts that helped us improve our manuscript! In case our answers have justifiably addressed your concerns, we respectfully thank you that support the acceptance of our work. As you know, your support holds great significance for us. Also, please let us know if you have any further questions. Look forward to further discussions!
>
> [1] Brandes, Nadav, et al. "ProteinBERT: a universal deep-learning model of protein sequence and function." Bioinformatics 38.8 (2022): 2102-2110.
>
> [2] Zhou, Hong-Yu, et al. "Protein Representation Learning via Knowledge Enhanced Primary Structure Modeling." bioRxiv (2023): 2023-01.
>
> [3] Zeming Lin, Halil Akin, Roshan Rao, Brian Hie, Zhongkai Zhu, Wenting Lu, Allan dos Santos Costa, Maryam Fazel-Zarandi, Tom Sercu, Sal Candido, et al. Language models of protein sequences at the scale of evolution enable accurate structure prediction. BioRxiv, 2022.
>
> [4] Hehe Fan, Zhangyang Wang, Yi Yang, and Mohan Kankanhalli. Continuous-discrete convolution for geometry-sequence modeling in proteins. In The Eleventh International Conference on Learning Representations, 2023.
>
> [5] Zuobai Zhang, Minghao Xu, Arian Jamasb, Vijil Chenthamarakshan, Aurelie Lozano, Payel Das, and Jian Tang. Protein representation learning by geometric structure pretraining. In International Conference on Learning Representations, 2023.
>
> [6] Varadi, Mihaly, et al. "AlphaFold Protein Structure Database: massively expanding the structural coverage of protein-sequence space with high-accuracy models." Nucleic acids research 50.D1 (2022): D439-D444.
>
> [7] John Ingraham, Vikas Garg, Regina Barzilay, and Tommi Jaakkola. Generative models for graph-based protein design. Advances in neural information processing systems, 32, 2019.
>
> [8] Anishchenko, Ivan, et al. "De novo protein design by deep network hallucination." Nature 600.7889 (2021): 547-552.
>
> [9] Torres, M., Yang, H., Romero, A. E. & Paccanaro, A. Protein function prediction for newly sequenced organisms. Nat. Mach. Intell. 3, 1050–1060 (2021).
>
> [10] Ibtehaz, Nabil, Yuki Kagaya, and Daisuke Kihara. "Domain-PFP allows protein function prediction using function-aware domain embedding representations." Communications Biology 6.1 (2023): 1103.
>
> [11] Zeming Lin, et al. Language models of protein sequences at the scale of evolution enable accurate structure prediction. BioRxiv, 2022.
>
> [12] Asgari, E. and Mofrad, M. R. K. (2015). Continuous distributed representation of biological sequences for deep proteomics and genomics. PLOS ONE.
>
> [13] Yang, K. K., Wu, Z., Bedbrook, C. N., and Arnold, F. H. (2018). Learned protein embeddings for machine learning. Bioinformatics.
>
> [14] Ferruz, N. and Höcker, B. (2022). Controllable protein design with language models. Nature Machine Intelligence, pages 1–12.
>
> [15] Oliver Johnson. Information theory and the central limit theorem. World Scientific, 2004.
>
> [16] Samy Bengio, et al. 2010. Label embedding trees for large multi-class tasks. Advances in Neural Information Processing Systems 23 (2010).
>
> [17] Xu Sun, Bingzhen Wei, Xuancheng Ren, and Shuming Ma. 2017. Label embedding network: Learning label representation for soft training of deep networks. arXiv preprint arXiv:1710.10393 (2017)
>
> [18] Wang, Zeyuan, et al. "Multi-level Protein Structure Pre-training via Prompt Learning." The Eleventh International Conference on Learning Representations. 2022.
>
> [19] Hu, Fan, et al. "A Multimodal Protein Representation Framework for Quantifying Transferability Across Biochemical Downstream Tasks." Advanced Science (2023): 2301223.

---

> > ### Comment · Reviewer_KkPB · 2023-11-21
> > **Response to Q12, and appendix D**
> >
> > I find the writing here is still quite confusing.  Generalization bound arguments are typically made with respect to a given model on different datasets.  This application here is to try and bound the loss of the student by the loss of the teacher network plus some distributional alignment terms of the student and teacher model latent variables.  But the notation is overloaded here in confusing ways:
> > - What does $Y$ specifically refer to here?  Is it shared by both models?  I did not think the student model had any notion of labeled data necessarily...
> > - mutual information is defined on random variables, not on their realizations; so equation (15) and it's description is very confusing for me
> >
> > On the whole, I think this is evidence that my original criticism of their being too many elements incorporated into this method without a cohesive story motivating their inclusion stands.  I acknowledge the hard work of the authors in other respects, and will increase my score accordingly, but I don't think the authors succeeded yet in producing the clearest version of this paper.

---

> ### Author Response · Authors · 2023-11-22
>
> **(4)** Regarding the authors answer to Q8 ('isotropic'), this description that the authors provide I think is the clearest description of what the
>  parameter controls, and maybe should be what appears in the text. Regarding the choice to incorporate functional annotations through label encoding rather than as a separate loss function, I appreciate the justification via Bengio et al and Sun et al. I think this justification should also appear in the text (and please add both citations to the manuscript).
>
> **A4** Thanks for your suggestions! We have added these contents to the revised manuscript.
>
> **(5)** I find the writing here is still quite confusing. Generalization bound arguments are typically made with respect to a given model on different datasets. This application here is to try and bound the loss of the student by the loss of the teacher network plus some distributional alignment terms of the student and teacher model latent variables. But the notation is overloaded here in confusing ways: (a) What does $Y$ specifically refer to here? Is it shared by both models? I did not think the student model had any notion of labeled data necessarily. (b) Mutual information is defined on random variables, not on their realizations; so equation (15) and it's description is very confusing for me
>
> **A5** Thanks for your valuable reviews! The teacher and student models can be used in different datasets. To reduce the generalization bound, we focus on optimizing marginal misalignment and obtain the derived loss for the student model, which is Eq. 9. (a) $y$ represents the underlying functional label for the student model. Although the student model may not have these functional labels, but we can assume that they exist for theoretical reasons. In Appendix D, we have mentioned we assume the source and target domains have the same support set (training set) to derive this theory, i.e,  the difference is that the teacher has functional labels, while the student does not. The derived misalignment Eq. 8 and the derived loss Eq. 9 are based on the assumption that the source and target domains have the same support set. Thus, the loss of Eq. 9 can be used in an unsupervised way for the student to predict functions. However, the student model is applied to different downstream tasks, like classification, which have classification classes. Thus, we add the supervised student loss $L_\mathrm{student}$ and the knowledge distillation loss the $L_{kd}$ as the final hybrid loss for the student to improve the performance on the classification task. (b) Mutual information is typically used to compute the relationship between random variables. However, if we have a way to discretize or represent the non-variable or actual values as random variables, we may be able to apply mutual information to measure their relationship. This would involve defining appropriate probability distributions for the variables and estimating the probabilities based on the observed data. In Eq. 15, the variable z represents the latent embedding, which is a learned representation of the input graph G. The variable y represents the data label or target variable, which provides information about the class or category to which the input data belongs. The variable G represents the input data itself. G is typically considered as the independent variable in the context of the equation. The mutual information term, denoted as I_S(·, ·), is calculated on the source domain. It quantifies the amount of information shared between the variables z and y (or G and y) in the source domain. It measures the dependence or correlation between these variables in the context of the source domain data.
>
>
>
> Thank you again for all the efforts that helped us improve our manuscript! In case our answers have justifiably addressed your concerns, we respectfully thank you that support the acceptance of our work. As you know, your support holds great significance for us. Also, please let us know if you have any further questions. Look forward to further discussions!
>
>
> [1] Ibtehaz, Nabil, Yuki Kagaya, and Daisuke Kihara. "Domain-PFP allows protein function prediction using function-aware domain embedding representations." Communications Biology 6.1 (2023): 1103.

---

### Author Response · Authors · 2023-11-20
**Global Response**

First and foremost, we would like to express our sincere gratitude for the insightful and constructive feedback provided by the reviewers on our manuscript. We greatly appreciate their positive reception of ProteinSSA's potential and its timely relevance in the field of protein research.

We are particularly thankful for Reviewer KkPB's recognition of the motivations and ideas behind our study, which aim to address the data imbalance in annotated protein data through distillation. We also appreciate their acknowledgment of the comprehensive protein sequential and structural features incorporated into the model's input, as well as the promising results achieved in fold classification. All reviewers have acknowledged the promising quality and significance of our work. Additionally, Reviewer G223's comments on the significance and originality of our research in the current scientific landscape, along with their extensive discussions on the problem definition and related work, are highly valued. Both Reviewer KkPB and Reviewer G223 have expressed acknowledgment of the ablation study, which we are encouraged.

We are grateful for the feedback received, particularly regarding the clarity of Section 3. We acknowledge that the writing may have been confusing for readers and have taken these concerns into account during the revision process. We value the insights provided by the reviewers and have diligently incorporated the suggestions into the revised draft.

Once again, we sincerely appreciate the reviewers' feedback and remain committed to continuously improving our research and manuscript based on their valuable insights. Thank you again!

---

### Author Response · Authors · 2023-11-23
**Global Response**

First and foremost, we would like to express our sincere gratitude for the insightful and constructive feedback provided by the reviewers on our manuscript, and we are moved, motivated and learned a lot from it.

Here, we want to emphasize the points of the derived equations and domain adaption theory again, which have been presented in the manuscript. There exists a misalignment for KL-guided domain adaptation method in the representation space between the source domain and the target domain because of data diversities. Theoretically, the student model should have a lower generalization bound when transferring knowledge from the teacher model (Eq.8). We reduce the misalignment and get the KL divergence loss. However, the derived equations are based on an assumption, i.e., the source and target domains have the same support set (training set), the difference is that the teacher has functional labels, while the student does not in reality, but we can deem the student model has underlying functional labels $y$. In this condition, the derived student loss (Eq.9), which approximates the ideal loss, can be used in an unsupervised way for the student model. However, we apply the student model in different downstream tasks, which have supervised labels. Thus, we combine the supervised student loss $L_\mathrm{student}$ and the KL divergence loss the $L_{kd}$ as the final hybrid loss (Eq.10) for the student to improve its performance on supervised tasks.

Once again, we sincerely appreciate the reviewers' feedback and remain committed to continuously improving our research based on their valuable insights. Thanks again!

---

### Meta-Review · Area_Chair_Jhqj · 2023-12-08

**Metareview:**

This paper proposes a multimodal protein representation framework using sequence, predicted structure and gene ontology (GO) annotations and shows that it enhances fold classification and function prediction. The paper's strength is new proposals for protein representations. The weakness is the lack of clarity of the suggested approach. It appears overly complex. The practical interest for biologists is also limited: fold classification is perhaps not so interest in the age of AlphaFoldDB and FoldSeek and EC numbers are not specific enough to learn about detailed function.

So in conclusion, multimodality is an interesting timely angle, but the current model and choice of downstream tasks could be improved. Therefore, rejection is recommended with a strong commendation to use the feedback to rework the framework.

**Justification For Why Not Higher Score:**

Lack of clarity and benchmarking not so interesting.

**Justification For Why Not Lower Score:**

None.

---

### Decision · Program_Chairs · 2024-01-16

Reject